# A Closer Look at Model Adaptation using Feature Distortion and Simplicity Bias

**Puja Trivedi**
CSE Department
University of Michigan
pujat@umich.edu

**Danai Koutra**
CSE Department
University of Michigan
dkoutra@umich.edu

**Jayaraman J. Thiagarajan**
Center of Applied Scientific Computing
Lawrence Livermore Natl. Laboratory
jjayaram@llnl.gov

## Abstract

Advances in the expressivity of pretrained models have increased interest in the design of adaptation protocols which enable safe *and* effective transfer learning. Going beyond conventional linear probing (LP) and fine tuning (FT) strategies, protocols that can effectively control feature distortion, i.e., the failure to update features orthogonal to the in-distribution, have been found to achieve improved out-of-distribution generalization (OOD). In order to limit this distortion, the LP+FT protocol, which first learns a linear probe and then uses this initialization for subsequent FT, was proposed. However, in this paper, we find when adaptation protocols (LP, FT, LP+FT) are also evaluated on a variety of safety objectives (e.g., calibration, robustness, etc.), a complementary perspective to feature distortion is helpful to explain protocol behavior. To this end, we study the susceptibility of protocols to simplicity bias (SB), i.e. the well-known propensity of deep neural networks to rely upon simple features, as SB has recently been shown to underlie several problems in robust generalization. Using a synthetic dataset, we demonstrate the susceptibility of existing protocols to SB. Given the strong effectiveness of LP+FT, we then propose modified linear probes that help mitigate SB, and lead to better initializations for subsequent FT. We verify the effectiveness of the proposed LP+FT variants for decreasing SB in a controlled setting, and their ability to improve OOD generalization and safety on three adaptation datasets.

## 1 Introduction

Through the use of larger datasets (Yalniz et al., 2019), better architectures (Zhai et al., 2022; Chen et al., 2022; Steiner et al., 2022; Tolstikhin et al., 2021), and different self-supervised learning (SSL) approaches (He et al., 2020; Chen et al., 2020; Grill et al., 2020; Caron et al., 2020), the quality of pretrained representations available for transfer learning tasks has dramatically improved. Indeed, representations from such high-quality SSL models have been found to be more robust (Hendrycks et al., 2019; Liu et al., 2021), transferable (Ericsson et al., 2021) and semantically consistent (Caron et al., 2021) than their supervised counterparts. In this regard, there is growing need for adaptation protocols that explicitly capitalize on these improved pretrained features to induce similar beneficial properties, *e.g.,* inducing more than just high accuracy on the target task, after models have been trained on the downstream task.

However, standard adaptation protocols that rely upon fine-tuning (FT) all model parameters or training only a linear probe (LP) while freezing the network parameters do not maximize the potential of high-quality representations. For example, while high-quality, pre-trained models have sufficiently expressive features to perform well on

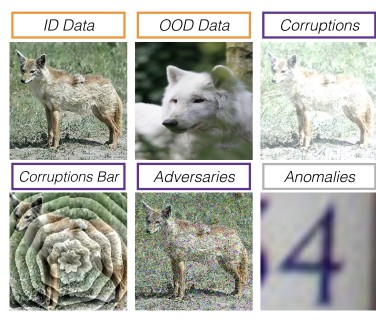

Figure 1: **Strong and Safe Adaptation.** Practical deployment in high risk applications requires that adapted models not only generalize well to in- and out-of distribution data of the downstream task, but they do so safely.

---

Correspondence to pujat@umich.edu.

both in-distribution (ID) and out-of-distribution (OOD) data, `LP` and `FT` are not able to effectively induce this property in adapted models (Andreassen et al., 2021). Recently, however, Kumar et al. (2022) proved that by modifying features only in the ID representation subspace, `FT` can lead to higher OOD error as it distorts directions outside the ID subspace that are needed for OOD generalization. As both ID and OOD subspaces are represented by the pretrained model, Kumar et al. demonstrate that limiting *feature distortion*, or controlling updates towards the ID subspace, can lead to improved ID and OOD performance. To this end, they propose a new protocol which performs `LP` prior to `FT` (abbrev. `LP + FT`). By first performing `LP`, this two-step process ensures that subsequent `FT` will remain in the vicinity of the original `LP` solution. This reduces the overall distortion towards the ID distribution subspace and improves performance.

While strong ID and OOD generalization on the target task is indeed an important aspect of transfer learning, practical, high-risk applications require that models are also safe (Hendrycks et al., 2021). For example, adapted models should also be well-calibrated, robust to corruptions or adversaries and able to reliably detect anomalous samples (see Figure 1). Given that existing adaptation protocols are primarily focused on improving generalization, it is unclear how existing protocols utilize high-quality pretrained features to promote safe adaptation, and if current protocol design perspectives, such as mitigating feature distortion, will also enable safe generalization.

**Our Work**: In this work, we seek to understand the factors relevant to the design of adaption protocols that promote effective *and* safe generalization. We take the first step towards this aim by (i) demonstrating limitations in existing `LP`, `FT`, and `LP+FT` protocols through an extensive, joint evaluation, and (ii) studying adaptation protocols through the complementary lens of avoiding simplicity bias, *i.e.*, the problematic tendency of deep neural networks (DNNs) to prefer simple, potentially brittle features over complex features (Soudry et al., 2018; Gunasekar et al., 2018; Geirhos et al., 2019; Hermann et al., 2020; Shah et al., 2020). Using the insights from our analysis, we propose three variants of the `LP+FT` protocol that jointly improve safety and generalization on three datasets. Our contributions can be summarized as follows:

- **Joint Analysis of Adaptation Protocol Safety and Generalization (Sec. 3).** We show that when adaptation protocols are evaluated with respect to both ID/OOD generalization and safety, `LP+FT` trails behind `LP` or `FT` on several safety metrics. This demonstrates that solely mitigating feature distortion may not be sufficient for safe generalization. We also observe that keeping subsequent `FT` close to `LP` solution is crucial for the improved OOD generalization of `LP+FT`. This motivates us to focus on improving the `LP` initialization as a mechanism for improving both safety and OOD performance.

- **Role of Simplicity Bias in (Unsafe) Adaptation (Sec. 4).** To understand how protocols may induce safe adaptation, we study how different protocols avoid simplicity bias. While simplicity bias (Shah et al., 2020; Geirhos et al., 2019) has been shown to underlie several problems in machine learning safety, to the best of our knowledge, we are the first to consider its role in adaptation settings. We demonstrate that protocols must not only reduce distortion, but also should mitigate simplicity bias for effective adaptation.

- **Improved Protocols for Mitigating Simplicity Bias and Distortion (Sec. 5).** We propose three, simple modified `LP+FT` protocols that help mitigate both simplicity bias and distortion (Sec. 4.1). In particular, we consider modifying the `LP` step with uncertainty-driven perturbations (Pagliardini et al., 2022), virtual adversarial training (Miyato et al., 2017) and model-soups (Wortsman et al., 2022), as they are simple and effective strategies. Across synthetic and real datasets, the modified protocols help improve safety and generalization to some extent.

## 2 RELATED WORK AND BACKGROUND

Here, we discuss the most relevant work on adaptation protocols and simplicity bias; we discuss additional related work in Sup. A.

**Adaptation Protocols.** For a comprehensive overview of transfer learning, please see the surveys of Zhuang et al. (2021) and Pan & Yang (2010). Here, we discuss the works that are most relevant to our own. Kirichenko et al. (2022) recently demonstrated that models are able to learn both core features and spurious features. However, classifiers can rely upon spurious features, harming

performance on minority groups. To reduce the reliance on spurious features, they propose to retrain the classifier on a small amount of "re-weighting" data, which allows the model to leverage the core features instead of the spurious features. Other modifications and heuristics have also been proposed to improve `FT`'s performance, including side-tuning (Zhang et al., 2019), which tunes a small secondary network that is then combined with the original model, using larger/smaller learning rates for the classifier, as well as regularization-based methods (Jiang et al., 2020). In this work, we focus on two popular and effective protocols, `LP` and `FT`. We additionally study the `LP+FT` protocol as it is theoretically-grounded, does not require re-weighting data, is designed to exploit high-quality pre-trained representations and achieves SOTA OOD performance during adaptation.

**Simplicity Bias.** It is well-known that DNNs demonstrate a bias toward simple, potentially less expressive features (Brutzkus et al., 2017; Soudry et al., 2018; Gunasekar et al., 2018; Geirhos et al., 2019; Hermann et al., 2020; Lubana et al., 2023), such as textures and backgrounds, and that this bias can lead to shortcuts that limit the generalization of DNNs. Indeed, recently Shah et al. (2020) formalized this intuition by more precisely defining simplicity bias, based on the number of linear components to define a decision boundary, and showed that SB leads to non-robust decision boundaries that affects a model's sensitivity to distribution shifts and adversarial perturbations. In brief, by learning simple features first, models become invariant to complex features, potentially leading to narrow decision boundaries which can fail to generalize under data shifts. Notably, DNNs exhibit this bias even when complex features are more expressive and necessary for fitting the distribution. While various techniques have recently been proposed to mitigate simplicity bias when training from scratch or in the context of pretraining (Teney et al., 2021), we are, to the best of our knowledge, the first to rigorously study the role of simplicity in the context of model adaptation.

## 3 JOINT ANALYSIS OF PROTOCOL SAFETY AND GENERALIZATION

In this section, we evaluate the performance of adaptation protocols across several additional safety objectives (Hendrycks et al., 2021), as practical transfer learning applications require both strong and safe generalization. Through this expanded evaluation, we find that no single protocol is optimal across all safety objectives. Indeed, the inability of `LP+FT` to induce safe adaptation indicates that a complementary perspective to feature distortion, namely simplicity bias, is necessary when designing generalizable and safe protocols (see Sec. 4). We further argue that by constraining models around the `LP` initialization during `FT`, `LP+FT` may inadvertently harm safety performance by hampering models' abilities to learn complex, task-specific features needed for robust generalization. While we expand upon the role of `LP` initialization in Secs. 4 and 5, we begin, here, by introducing the expanded evaluation and experimental setup.

**Experimental Setup.** Three downstream adaptation tasks (and their respective OOD distributions) are considered: CIFAR-10 (ID) → {STL10, CIFAR10.1} (OOD), Domainnet-Sketch → {Domainnet-ClipArt, Domainnet-Painting, Domainnet-Real} and Living17 (Source) → Living17 (Target). These datasets are selected as they correspond to two different types of distribution shifts (standard domain adaptation and subpopulation) and three levels of distortion (low, medium, high). A MoCo-V2 ResNet-50 (He et al., 2020) pretrained on ImageNet-1K is used as the base-feature extractor for CIFAR10 and Living17 experiments, and the CLIP ResNet-50 image encoder pretrained on 400 million (image,text) pairs is used for Domainnet-Sketch. These models are selected as they provide sufficiently high-quality representations capable of generalizing to both ID and OOD downstream data (Kumar et al., 2022). We perform grid-search to find the best hyper-parameters, and average over 3 seeds. See Sup. B.2 for additional details.

**Expanded Evaluation.** In addition to OOD accuracy on the aforementioned distribution shifts, we report performance on the following metrics in order to evaluate adapted models on key problems in machine learning safety (Hendrycks et al., 2021). Our evaluation setup is inspired by Hendrycks et al. (2022):

- *Mean Corruption Accuracy (mCA/m$\bar{C}$A):* We consider two sets of corruptions: the 15 naturalistic corruptions ($Corr$) (Hendrycks & Dietterich, 2019), and 10 perceptually dissimilar corruptions ($\overline{Corr}$) (Mintun et al., 2021). Corruptions are applied to the ID test dataset and the average accuracy over each set is reported.

- *Calibration Error (RMSE)*: It is important that models are well-calibrated so that practitioners may trust the provided predictions in high-risk applications Guo et al. (2017). We measure the root mean square error of calibration as follows: $\sqrt{\mathbb{E}_C\left[(\mathbb{P}(Y = \hat{Y} \mid C = c) - c)^2\right]}$, where $C$ indicates the confidence scores, while $\hat{Y}$ and $Y$ denote the model's predictions and ground-truth labels, respectively.

- *Anomaly Detection Performance (AUROC):* Recognizing when samples are anomalous allows models to abstain from making uninformed and inapplicable predictions. We consider samples from Blobs, Gaussian, LSUN, Places69, Rademacher, Textures, and SVHN datasets as anomalies and report the AUROC (area under the ROC curve) of the binary classification problem of detecting such samples as anomalies.

- *Adversarial Accuracy:* DNNs are well-known to be fooled by imperceptible distortions (Ilyas et al., 2019). We use a 2/225, 10-step PGD (Madry et al., 2018) attack to measure the robustness of models to such perturbations.

We make the following observations regarding the behavior of different protocols using this expanded evaluation. In brief, we find that no single protocol is effective across all datasets in jointly obtaining strong and safe adaptation, and that, on low distortion adaptation tasks, the quality of the LP initialization is critical as pre-trained feature extractor is not substantially updated during LP+FT.

OBSERVATION 1: MITIGATING FEATURE DISTORTION MAY NOT INDUCE SAFE ADAPTATION.

Here, we ask how protocols perform when we consider both safety and generalization objectives to better understand the feature distortion perspective. In particular, if LP+FT is able to outperform LP and FT in this expanded evaluation, then it suggests that solely mitigating feature distortion during FT may be sufficient to induce robust adaptation. To test this claim, we rank protocol performance for each safety metric, where ranks are first computed for each dataset separately, and then averaged. Results are shown in Fig. 2. Smaller ranks correspond to better performance.

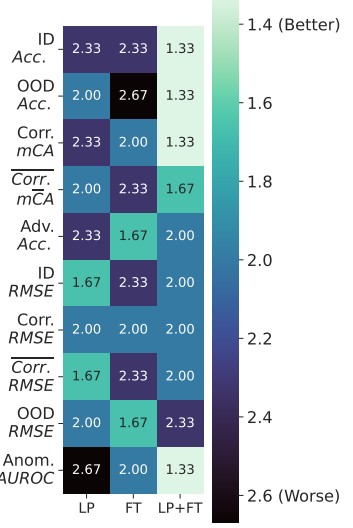

*Results.* LP+FT obtains the best rank for ID and OOD accuracy as expected, as well as $Corr$ and $\overline{Corr}$ accuracy. However, we also see that FT is better ranked for Adversarial Accuracy and OOD calibration, while LP is better ranked for ID calibration and $\overline{Corr}$ calibration. However, given that LP+FT trails behind protocols that are not explicitly designed to limit distortion on some safety metrics, it is clear that a complementary perspective is needed to better understand protocol behavior. Indeed, LP+FT has the best *average* rank, indicating that it is a good starting point to improve upon.

Figure 2: **Disparate Performance of Protocols.** We plot the average rank of each protocol for different safety and generalization metrics. We see no single protocol achieves top rank across all metrics.

The above results are aggregated across different types of distribution shifts; we extend this analysis next by considering the interplay between individual datasets and protocol performance. These detailed results are presented in Table 1.

OBSERVATION 2: LINEAR PROBING SOLUTIONS MATTER.

Naturally, the amount of distortion required to effectively adapt a pretrained model to a downstream task will vary in accordance to the similarity of the downstream and pretraining data. Here, we seek to understand how protocols behave under different levels of distortion. In particular, we hypothesize that the LP initialization becomes more influential for LP+FT in low distortion settings, as subsequent FT remains in the vicinity of initialization. To this end, we compute the batched centered kernel alignment (CKA) score (Nguyen et al., 2021) with respect to the adapted and pretrained models, and take a closer look at performance across metrics. We note that while CKA is better suited for measuring distortion than the L2 norm as used by Kumar et al. (2022), other neural representation metrics can also be used (Ding et al., 2021; Davari et al., 2023).

*Results.* As shown in Fig. 3, we see that minimal distortion (CKA $\geq$ 0.9) is required to obtain competitive LP+FT performance on DomainNet and Living17. However, on CIFAR10, which requires the most distortion as evidenced by lower CKA scores, FT is the most effective protocol for safety measures and is very comparable on generalization performance (see Table 1).

The effectiveness of LP and LP+FT on Living17 in improving OOD generalization over FT is hardly surprising, as Living17 is a subset of ImageNet, on which the base feature-encoder was already trained.

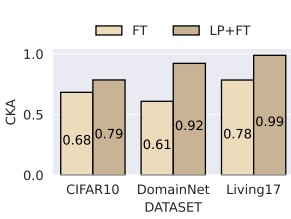

Figure 3: **Dataset Distortion.** We plot the CKA similarity between adapted and pretrained models. DomainNet and Living17 require low distortion, as seen by performance of LP+FT across metrics with high CKA (> 0.9).

In contrast, on DomainNet, the difficulty of FT in matching the ID *test* task performance, despite achieving high training accuracy, suggests FT may learn a solution that relies upon shortcuts (or simple features) that do not generalize. We emphasize that LP+FT greatly benefits from strong LP initializations on these low-distortion datasets as corresponding CKA scores show that very limited updates are made during FT. While LP+FT does induce meaningful improvements over LP on Living17 and performs comparably to LP on DomainNet, we stress the model must be kept close to the LP initialization during FT. Indeed, to obtain acceptable LP+FT performance, small learning rates (3e-7,1e-5) and frozen batch-norm parameters during FT are necessary.

*Summary.* Taken jointly, these results suggest that while solely mitigating feature distortion may not be sufficient to ensure that adapted models perform well on safety metrics across different levels of shift, improving the LP initialization may be a viable solution to obtaining strong and safe generalization. Indeed, the effectiveness of LP+FT on low distortion datasets and its high average ranking indicates that it is a promising protocol to build upon. To understand how to build better protocols, we next introduce *simplicity bias* as a complementary perspective to feature distortion.

| Dataset | Protocol | Generalization | | Robustness | | | Calibration | | | | Anomaly Det. | Rep. Similarity |
| | | ID | OOD[2] | C | $\overline{C}$[3]. | Adv. | ID | C | $\overline{C}$ | OOD. | Out-of-Class | ID |
| | | Acc. | Acc. | Acc. | Acc. | Acc. | 1-RMS | 1-RMS | 1-RMS | 1-RMS | AUROC | CKA |
|---|---|---|---|---|---|---|---|---|---|---|---|---|
| CIFAR10 | LP | 0.9138 | 0.8188/0.8192 | 0.6912 | 0.6553 | 0.0003 | 0.95945 | 0.83025 | 0.8142 | 0.8696 | 0.6206 | 1.0000 |
| CIFAR10 | FT | **0.9539** | **0.8962**/0.8545 | 0.7434 | 0.7553 | 0.0231 | **0.9668** | **0.83635** | **0.8453** | **0.9232** | **1.0000** | 0.6831 |
| CIFAR10 | LP+FT | 0.9442 | 0.8775/**0.8581** | 0.6921 | 0.6790 | 0.0018 | 0.9521 | 0.7849 | 0.7721 | 0.88633 | 0.6511 | 0.7853 |
| DomainNet | LP | 0.8913 | **0.8013** | 0.6019 | **0.6020** | 0.1768 | **0.9638** | **0.9045** | **0.8571** | **0.9264** | 0.8679 | 1.0000 |
| DomainNet | FT | 0.7613 | 0.4522 | 0.5186 | 0.2744 | **0.4164** | 0.8368 | 0.7234 | 0.7234 | 0.6379 | 0.8841 | 0.6092 |
| DomainNet | LP+FT | **0.8985** | 0.7990 | **0.6343** | 0.5979 | 0.1927 | 0.9566 | 0.8445 | 0.8445 | 0.8899 | **0.9022** | 0.9222 |
| Living17 | LP | 0.9521 | 0.8124 | 0.7010 | 0.7377 | **0.2350** | 0.9313 | 0.8693 | 0.8801 | **0.9117** | 0.9907 | 1.0000 |
| Living17 | FT | 0.9518 | 0.7168 | 0.7011 | 0.7164 | 0.1563 | 0.8873 | 0.9019 | 0.8604 | 0.9295 | 0.9794 | 0.7847 |
| Living17 | LP+FT | **0.9643** | **0.8261** | **0.7426** | **0.7671** | 0.2135 | **0.9782** | **0.9472** | **0.9451** | 0.8742 | **0.9924** | 0.9887 |

Table 1: **Safety and Generalization Performance of Adaptation Protocols.** (Best in **bold**. Second best underlined.) While LP+FT indeed achieves strong ID and OOD performance across datasets, we see that different protocols may perform better when safety evaluation is also considered. For CIFAR-10, which requires the most distortion as evidenced by lower CKA scores, we see that FT is the most effective; LP+FT and LP are most effective, respectively, on Living17 and DomainNet, which require significantly less distortion. This suggests that, while mitigating feature distortion is effective for improving generalization, it is not always sufficient for also improving safety.

# 4 MITIGATING SIMPLICITY BIAS & FEATURE DISTORTION FOR SAFE ADAPTATION

As discussed in Sec. 2, simplicity bias (SB) underlies various safety issues in machine learning as models may learn to rely upon simple features that often do not generalize under distribution shifts, such as corruptions or adversaries (Shah et al., 2020). Therefore, we argue that mitigating feature distortion in a way that minimizes this bias can be a valid mechanism to improve both safety and generalization performance. Correspondingly, in this section, we first measure the propensity of different protocols to simplicity bias in a controlled setting. In particular, given our previous observation that LP+FT models remain in close vicinity of the LP solution after FT, we

focus on improving the performance of this initial `LP` initialization so that we may capitalize upon `LP+FT` strong OOD performance, while simultaneously improving safety. To this end, we propose three light-weight `LP+FT` variants that are able to both reduce distortion and SB. We begin by introducing our synthetic dataset and experimental setup.

**Dataset.**    As shown in Fig. 4, we create "dominoes" of complex and simple features by pairing each class (Shah et al., 2020) from CIFAR10 (complex) with the corresponding "digit" class in MNIST (simple), e.g., "bird" samples are paired with digit "2" samples, where the label for each domino is determined by the complex, CIFAR10 sample. Datasets with three levels of correlation (95%, 99%, 100%) between the simple and complex features are constructed for training. While 100% correlation allows models to only learn the simple feature for perfect generalization, the more realistic lower correlation settings require models learn at least some aspect of the complex features.

**Experimental Setup.**    For evaluation, we also construct a randomized (10% correlation) variant, where simple features are randomly paired with complex features. We give two examples in panels 3 and 4 of Fig. 4. To assess OOD generalization, we create a variant where complex features are sampled from STL10, instead of CIFAR10, e.g., panels 1 and 2 in Fig. 4.

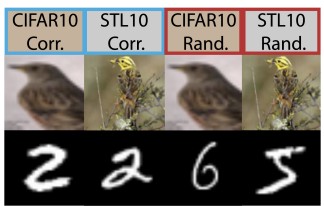

*Metrics.*  We assess the reliance on simple features using the following metrics: (1) *Randomized Accuracy*: the accuracy on the variant where samples contain random pairings between simple and complex features; (2) *Correlated Accuracy*: accuracy when pairings between simple and complex features remain correlated. Models that *are* susceptible to simplicity bias will have *high* Correlated Accuracy and *low* Randomized Accuracy. Likewise, models that are *not* susceptible to simplicity bias will have relatively *lower* correlated accuracy and *higher* randomized accuracy.

Figure 4: **Synthetic Data with Simple and Complex Features.** Using a synthetic dominoes dataset (Shah et al., 2020), we study the effect of simplicity bias on safety and OOD generalization.

*Training Details.* A MoCo-V2 ResNet-50 (He et al., 2020) pretrained on ImageNet-1K is the base-feature extractor. See Supp. B.2 for additional details. We performed grid-search to find the best parameters. Results are over 3 seeds and 3 correlation strengths.

| Protocol | *Correlation = 0.95* | | | *Correlation = 0.99* | | | *Correlation = 1.00* | | |
|---|---|---|---|---|---|---|---|---|---|
| | Corr. ID Acc. | Corr. OOD Acc. | Rand.OOD Acc. | Corr. ID Acc. | Corr. OOD Acc. | Rand.OOD Acc. | Corr. ID Acc. | Corr. OOD Acc. | Rand.OOD Acc. |
| LP | 0.9728 | 0.9156 | 0.7910 | 0.9809 | 0.9386 | **0.7862** | 0.9836 | 0.9505 | **0.7696** |
| FT | 0.9866 | **0.9814** | 0.6021 | **0.9923** | **0.9928** | 0.3629 | 0.9855 | **0.9789** | 0.2697 |
| LP+FT | **0.9902** | 0.9793 | **0.8422** | 0.9844 | 0.9484 | 0.7813 | 0.9874 | 0.9594 | 0.7548 |
| FT (Scratch) | 0.9557 | 0.9123 | 0.1820 | 0.9861 | 0.9595 | 0.1201 | **0.9952** | 0.9444 | 0.1055 |

Table 2: **Simplicity Bias and Performance of Adaptation Protocols.** Using the synthetic dominoes dataset, we measure the propensity of different models to simplicity bias by measuring the Corr. OOD and Rand. OOD accuracy With highest Corr. OOD accuracy and lowest Rand. OOD accuracy, we see that `FT` is particularly susceptible to inducing simplicity bias.

*Results.*  Given the above experimental setup, we report the performance of different adaptation protocols in Table. 2. Across all correlation strengths, `FT` has the lowest *Rand.* OOD accuracy and high *Corr.* OOD accuracy. This clearly indicates that `FT` has learned to rely upon simple features, effectively disregarding the expressive features of the pretrained model, and is easily susceptible to simplicity bias. In contrast, by preserving the expressive features of the underlying feature encoder, `LP` best mitigates simplicity bias in high correlation (0.99,1.0) settings as evidenced by the highest *Rand* OOD accuracy (though *Corr.* ID/OOD accuracy does slightly suffer). However, in moderate correlation (0.95), `LP+FT` improves upon `LP` to achieve better *Corr.* OOD accuracy than `LP` and the best *Rand.* OOD accuracy across protocols. This suggests that when the correlation is not extreme, that moderate distortion, given a suitable `LP`, is in fact beneficial to mitigating simplicity bias. At higher correlation strengths (0.99,1.0), however, `LP+FT` has lower *Rand* OOD accuracy, while improving the *Corr.* OOD accuracy relative to `LP`, indicating in such extreme settings the distortion incurred by subsequent `FT` is not beneficial and has increased reliance upon simple features.

## 4.1 IMPROVED LINEAR PROBES FOR MITIGATING SIMPLICITY BIAS

As discussed earlier, adaptation protocols have varying susceptibility to simplicity bias, and mitigating this susceptibility can help improve generalization and safety. In particular, we observe that `LP+FT` and `LP` are effective protocols for reducing reliance upon simple features on both synthetic datasets, and on low-distortion real datasets (Sec. 3). However, as some level of distortion is typically required when adapting to downstream tasks to obtain sufficient ID task performance, we propose new variants of the `LP+FT` protocol that attempt to enable the subsequent `FT` step to distort features without compromising generalization or increasing simplicity bias.

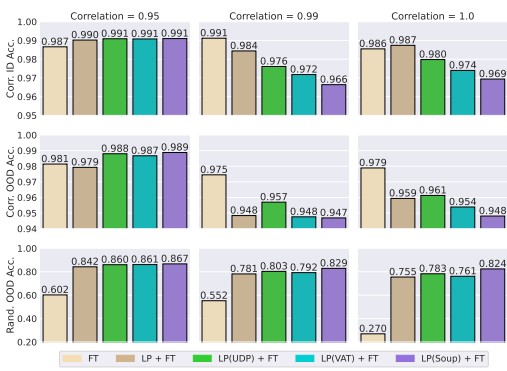

We note that, while it is possible to modify the `FT` step as well, modifications to `LP` are inexpensive as the feature-encoder is not updated. Moreover, as discussed in Sec. 3, fine-tuned solutions remain in close vicinity of initial `LP` initializations, further motivating strong starting solutions. To this end, we introduce the following modifications to the `LP` step of `LP+FT` below, where $h$ are the hidden representations, $\theta$ model parameters, $y$ labels, $\hat{y}$ predicted classes, $C$ the number of classes, $g$ the classifier, and $\delta$

Figure 5: **Hardness Promoting Augmentations help Mitigate Simplicity Bias.** We evaluate the modified `LP+FT` protocols on the dominoes dataset, and find they improve the Rand. OOD Accuracy over vanilla `FT` and `LP+FT`. This suggests that modified protocols can rely less upon shortcuts or simple features.

the perturbation. See Supp. B.1 for additional discussion on the choice of these mitigation strategies and Supp. B.2 for discussion on the importance of applying mitigations during `LP`.

- **LP(VAT)**: Virtual adversarial training (VAT) (Miyato et al., 2017) enforces local distribution smoothness by minimizing the KL-divergence between the predictions of perturbed pairs of examples. Since we are using expressive pretrained models, such perturbations may be meaningful in the inverted latent space as well, and resulting classifiers become robust in some $\epsilon$-neighborhood around each latent-space input. Formally, let $\epsilon$ be some perturbation budget, and $\alpha$ a hyper-parameter weighting distributional label smoothness, we minimize the following loss: $\min_\theta \mathcal{L}_{\text{CE}}(g_\theta(h), y) - \alpha \text{KL}\left[p\left(y \mid g_\theta(h)\right), p\left(y \mid g_\theta(h + \delta)\right)\right]$ where $\delta := \arg\max_{\|\delta\|_2 \leq \epsilon} \text{KL}\left[p\left(y \mid g_\theta(h)\right), p\left(y \mid g_\theta(h + \delta)\right)\right]$.

- **LP(UDP)**: Instead of maximizing the loss, uncertainty-driven perturbations (UDP) (Pagliardini et al., 2022) adversarially maximize a model's estimated uncertainty. UDPs have been shown to be effective in decreasing simplicity bias and improving generalization in non-adaptation settings. Formally, they can be defined as: $\delta_u = \arg\max_{\|\delta\|_2 \leq \epsilon} \mathcal{H}(g_\theta(h) + \delta)$, where $\mathcal{H}(g_\theta(h)) = -\sum_{c \in C} \hat{y}_c \log \hat{y}_c$, (e.g., entropy of predictions).

- **LP(Soup)**: Inspired by Wortsman et al. (2022), we train multiple, sparse, linear probes jointly and then take the average of their weights (aka soup) as the learned `LP` for subsequent `FT`. While soups of large models improve generalization by combining models from the same low-error basin, we consider sparse classifiers soups as an alternative strategy which seeks to average diverse decision rules, to avoid relying upon a single set of simple features. Formally, given $k$ classifiers, we minimize $\min_{\theta_1 \ldots k} \frac{1}{k} \sum_{i=1}^{k} \mathcal{L}_{\text{CE}}(g_{\theta_i}(h), y)$ and let $g_{\bar{\theta}} = \frac{1}{k} \sum_{i}^{k} \theta_i$ for the `FT` step.

**Empirical Evaluation of Hardness Promoting Augmentations.** We evaluate the effectiveness of the above `LP` variants, which we collectively refer to as "hardness-promoting", in mitigating the simplicity of bias of `LP+FT`. We make the following observations (see Fig. 5).

Across all correlation strengths, we find that using the modified hardness-promoting `LP`s during `LP+FT` (aka hp-`LP+FT`) improves the Rand. OOD Accuracy over vanilla `LP+FT`($\geq 2\%$) and `FT`($> 20\%$). This clearly indicates that hp-`LP+FT` is indeed effective in decreasing reliance on

| Protocol | Generalization | | Robustness | | | Calibration | | | | Anomaly Det. | Rep. Similarity |
|---|---|---|---|---|---|---|---|---|---|---|---|
| | ID | OOD | C | $\overline{\text{C}}$ | Adv. | ID | C | $\overline{\text{C}}$ | OOD. | Out-of-Class | ID |
| | Acc. | Acc. | Acc. | Acc. | Acc. | 1-RMS | 1-RMS | 1-RMS | 1-RMS | AUROC | CKA |
| LP | 0.9521 | 0.8124 | 0.7010 | 0.7378 | 0.2350 | 0.9313 | 0.8693 | 0.8802 | 0.9117 | 0.9907 | 1.0000 |
| FT | 0.9518 | 0.7168 | 0.7011 | 0.7164 | 0.1563 | 0.8873 | 0.9019 | 0.8604 | **0.9295** | 0.9794 | 0.7847 |
| LP+FT | 0.9643 | 0.8261 | 0.7426 | 0.7671 | 0.2135 | **0.9782** | 0.9472 | 0.9451 | 0.8742 | 0.9924 | 0.9887 |
| LP(UDP) | 0.9524 | 0.8110 | 0.7017 | 0.7382 | 0.2353 | 0.9308 | 0.8691 | 0.8801 | 0.9118 | 0.9908 | 1.0000 |
| LP(VAT) | 0.9524 | 0.8122 | 0.7010 | 0.7379 | 0.2345 | 0.9299 | 0.8682 | 0.8791 | 0.9103 | 0.9907 | 1.0000 |
| LP(Soup) | 0.9439 | 0.7996 | 0.6874 | 0.7290 | **0.2451** | 0.8806 | 0.7868 | 0.8094 | 0.9064 | 0.9897 | 1.0000 |
| LP(UDP)+FT | 0.9637 | **0.8265** | 0.7448 | 0.7681 | 0.2157 | 0.9768 | 0.9464 | 0.9467 | 0.8757 | 0.9927 | 0.98927 |
| LP(VAT)+FT | **0.9647** | 0.8247 | 0.7425 | 0.7650 | 0.2224 | 0.9727 | **0.9521** | 0.9463 | 0.8775 | 0.9925 | 0.9893 |
| LP(Soup)+FT | 0.9608 | 0.8163 | **0.7456** | **0.7684** | 0.1855 | 0.9760 | 0.9498 | **0.9492** | 0.8678 | **0.9936** | 0.98540 |

Table 3: **Living17: Hardness Promoting Augmentation and Adaptation.** In this low-distortion adaptation setting, we see that vanilla LP+FT is an effective baseline and that hardness promoting variants of LP+FT tend to perform comparably.

simple features, potentially also leading to improved safety. Furthermore, with the exception of LP(Soup)+FT, hp-LP+FT also performs better than vanilla LP+FT on Corr. OOD accuracy. Vanilla FT does outperform all LP+FT protocols in this setting, but this is due to reliance upon simple features. Lastly, we observe that with respect to Corr. ID Accuracy that hp-LP+FT improves performance at low correlation strength, but slightly loses performance at higher correlation strengths. This is not entirely unexpected as FT's reliance upon simple features will be useful in the correlated setting. Given that hp-LP+FT is able to reduce reliance upon simple features in this controlled setting, we next evaluate whether these modified protocols are beneficial in improving the performance of LP+FT on real datasets.

## 5 EVALUATING GENERALIZATION AND SAFETY OF THE LP+FT FAMILY

Given the effectiveness of incorporating hardness promoting (hp) augmentations with the family of LP+FT protocols (hp-LP+FT) in mitigating simplicity bias in a synthetic setting, we further evaluate the modified protocols on the three real-world datasets (Living17, DomainNet, and CIFAR10) with respect to the generalization and safety metrics introduced in Sec. 3. We present our results in Tables 3,4, and 5); our observations are summarized below. Any method-specific hyperparameters (e.g., epsilon) are tuned using ID validation data and all results are reported over three seeds. We provide additional results in Supp. C.

*Results.* As discussed in Sec. 3, these three datasets represent scenarios where different levels of distortion are necessary when adapting the pretrained model. On Living17, a setting which requires minimal distortion during adaptation, we see that vanilla LP+FT is quite effective with respect to both generalization and safety metrics and is a difficult baseline to surpass. Indeed, while hp-LP+FT variants do not lead to significant benefits, they generally perform comparably to vanilla LP+FT. On DomainNet, a setting where fairly low distortion is required for LP+FT but FT struggles to find a good solution, we see that hp-LP+FT variants induce some slight benefits with respect to ID/OOD generalization and robustness, though vanilla LP and hp-LP have better calibration performance. In contrast on CIFAR10, which requires more distortion to obtain an acceptable solution, we see that hp-LP+FT variants lead to improved generalization and a noticeable boost in corruption robustness. LP(VAT)+FT and LP(VAT) are particularly effective in this regard. Lastly, across all datasets, we observe that hp-LP+FT protocols lead to similar distortion to vanilla LP+FT, which suggests that any additional benefits of hp-LP+FT should not be attributed to only reducing feature distortion.

*Discussion.* We find that while vanilla LP+FT is already an effective protocol, especially in settings where low distortion is required, hp-LP+FT can provide some benefits and performs competitively. We suspect that the performance of these modified protocols can further be improved if more sophisticated simplicity bias mitigation strategies are used. Indeed, our central claim, that adaptation protocols should mitigate feature distortion and simplicity, is not dependent on a specific strategy. We additionally note that while such mitigation strategies may optionally *also* be used during FT, they cannot *solely* be used in FT. Indeed, in the case of extreme simplicity, if the LP classifier relies upon simple features to find a low-loss solution, during the subsequent FT step, gradients may not be

| Protocol | Generalization | | Robustness | | | Calibration | | | | Anomaly Det. | Rep. Similarity |
|---|---|---|---|---|---|---|---|---|---|---|---|
| | ID | OOD | C | $\overline{C}$ | Adv. | ID | C | $\overline{C}$ | OOD. | Out-of-Class | ID |
| | Acc. | Acc. | Acc. | Acc. | Acc. | 1-RMS | 1-RMS | 1-RMS | 1-RMS | AUROC | CKA |
| LP | 0.8913 | 0.8013 | 0.6019 | 0.6020 | 0.1768 | 0.9638 | 0.9264 | 0.9045 | 0.9014 | 0.8679 | 1.0000 |
| FT | 0.7613 | 0.4522 | 0.5186 | 0.2744 | 0.4164 | 0.8368 | 0.7234 | 0.7234 | 0.6379 | 0.8841 | 0.6092 |
| LP+FT | 0.8985 | 0.7990 | 0.6343 | 0.5979 | 0.1927 | 0.9566 | 0.8445 | 0.8445 | 0.8899 | 0.9022 | 0.9222 |
| LP(UDP) | 0.8919 | **0.8021** | 0.6022 | 0.6101 | 0.1345 | 0.9635 | 0.9250 | **0.9047** | 0.8619 | 0.8714 | 1.0000 |
| LP(VAT) | 0.8836 | 0.7914 | 0.5893 | 0.5963 | 0.1687 | 0.8897 | **0.9552** | 0.8905 | **0.9178** | 0.8735 | 1.0000 |
| LP(Soup) | 0.8787 | 0.7977 | 0.5951 | 0.6048 | 0.1731 | 0.8844 | 0.9479 | 0.8861 | 0.9176 | 0.8661 | 1.0000 |
| LP(UDP)+FT | 0.9033 | 0.7965 | 0.6414 | **0.6178** | 0.1778 | 0.9436 | 0.8533 | 0.79415 | 0.752 | 0.8857 | 0.9662 |
| LP(VAT)+FT | 0.9048 | 0.8009 | **0.6466** | 0.6131 | 0.1942 | **0.9686** | 0.8911 | 0.8428 | 0.7985 | **0.9204** | 0.9370 |
| LP(Soup)+FT | **0.9051** | 0.8013 | 0.6393 | 0.6091 | **0.1954** | 0.9670 | 0.9042 | 0.8692 | 0.8246 | 0.9097 | 0.9281 |

Table 4: **DomainNet: Hardness Promoting Augmentations and Adaptation.** While relatively low distortion is induced by `LP+FT`, `FT` struggles to find a viable solution. Here, hardness-promoting `LP+FT` variants, particularly `LP(VAT)+FT` do slightly improve ID and OOD generalization as well as robustness to corruptions.

| Protocol | Generalization | | Robustness | | | Calibration | | | | Anomaly Det. | Rep. Similarity |
|---|---|---|---|---|---|---|---|---|---|---|---|
| | ID | OOD | C | $\overline{C}$ | Adv. | ID | C | $\overline{C}$ | OOD. | Out-of-Class | ID |
| | Acc. | Acc. | Acc. | Acc. | Acc. | 1-RMS | 1-RMS | 1-RMS | 1-RMS | AUROC | CKA |
| LP | 0.9138 | 0.8190 | 0.6912 | 0.6553 | 0.0003 | 0.9595 | 0.8303 | 0.8142 | 0.8696 | 0.6206 | 1.0000 |
| FT | 0.9539 | 0.8754 | 0.7434 | **0.7553** | 0.0231 | 0.9668 | 0.8364 | 0.8453 | 0.9232 | **1.0000** | 0.6831 |
| LP+FT | 0.9442 | 0.8678 | 0.6921 | 0.6790 | 0.0018 | 0.9521 | 0.7849 | 0.7721 | 0.8864 | 0.6511 | 0.7853 |
| LP(UDP) | 0.9033 | 0.8356 | 0.6948 | 0.6643 | 0.0003 | **0.9689** | 0.9111 | 0.9023 | 0.9277 | 0.9033 | 1.0000 |
| LP(VAT) | 0.8977 | 0.8251 | 0.6742 | 0.6483 | 0.0002 | 0.9265 | **0.9255** | **0.9139** | 0.9375 | 0.7200 | 1.0000 |
| LP(Soup) | 0.9052 | 0.8353 | 0.6917 | 0.6588 | 0.0003 | 0.9605 | 0.9205 | 0.9037 | 0.9364 | 0.8859 | 1.0000 |
| LP(UDP)+FT | 0.944 | 0.8848 | 0.7028 | 0.6986 | 0.0004 | 0.9670 | 0.8472 | 0.8476 | 0.9237 | 0.9559 | 0.7764 |
| LP(VAT)+FT | **0.9611** | **0.8900** | **0.7442** | 0.7321 | 0.0027 | 0.9294 | 0.8355 | 0.8281 | 0.9178 | 0.8276 | 0.7839 |
| LP(Soup)+FT | 0.9466 | 0.8892 | 0.7031 | 0.6931 | 0.0001 | 0.9678 | 0.8390 | 0.8287 | 0.9216 | 0.9265 | 0.7806 |

Table 5: **CIFAR10: Hardness Promoting Augmentations and Adaptation.** In contrast to Living17 and DomainNet, `FT` is more effective than `LP+FT` in the safety metrics and performs comparably on ID/OOD generalization. However, hardness-promoting variants, particularly `LP(VAT)`, see noticeable improvements with respect to generalization & corruptions, performing comparably to `FT`.

back propagated in directions that contain complex features. This entails that the decision boundary continues to rely upon simple features and is at risk of reduced safety performance. We provide further discussion in Supp.B.2. To this end, we recommend incorporating hardness-promoting augmentations during `LP` as a potential safe-guard to simplicity bias.

## 6 CONCLUSION

In this paper, we took a closer look at the behavior of protocols designed for adapting large-scale pretrained models to downstream datasets. While it is argued that adaptation protocols should be designed to mitigate feature distortion (e.g., `LP+FT`) in order to improve ID and OOD generalization, we found that when additional aspects of safe generalization are evaluated (e.g., prediction calibration error, adversarial robustness etc.), mitigating feature distortion alone is not sufficient. We then considered the complementary perspective, that adaptation protocols should also mitigate simplicity bias. Using a synthetic dominoes dataset that allows for control over the correlation between simple and complex features, we found that protocols have varying levels of effectiveness in reducing reliance upon simple features. While, as expected, `FT`, is most susceptible to simplicity bias, we see that `LP+FT` is able to balance both distortion and simplicity bias in settings where the correlation between simple and complex features is not too extreme. Motivated by the benefits of `LP+FT` and given the known relationship between simplicity bias and sub-optimal generalization, we used "hardness-promoting" `LP` initializations (virtual adversarial, uncertainty-driven perturbations, sparse soups) to further improve `LP+FT`'s performance. These modifications helped reduce `LP+FT`'s reliance upon simple features on the synthetic dataset. On three real-world datasets, these modified protocols led to some improvements in safety and generalization performance, further validating the need to consider both distortion and simplicity bias when designing adaptation protocols.

ACKNOWLEDGMENTS

We thank Ekdeep Singh Lubana for several helpful discussions during the course of this project. This work was performed under the auspices of the U.S. Department of Energy by the Lawrence Livermore National Laboratory under Contract No. DE-AC52-07NA27344, Lawrence Livermore National Security, LLC.and was supported by the LLNL-LDRD Program under Project No. 21-ERD-012. It was also partially supported by the National Science Foundation under CAREER Grant No. IIS 1845491. PT began this work as an intern at Lawrence Livermore National Laboratory.

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

# APPENDIX

## A    ADDITIONAL RELATED WORK

For a comprehensive overview of transfer learning, please see the surveys of Zhuang et al. and Pan & Yang. Here, we discuss a few directly works directly relevant to our own.

Recently, Kumar et al. demonstrated that learning probing prior to fine-tuning (e.g., `LP+FT`) can improve both in-distribution and out-of-distribution performance when transferring to a downstream task given a highly expressive, pretrained model. They demonstrated that `FT` only modifies features in the ID representation subspace and not in other directions, which can lead higher OOD error as direction outside the ID subspace are necessary for OOD generalization. However, by initializing `FT` with a trained linear probe, feature distortion can be decreased since this initialization is closer to optimal model, and thus requires less distortion in ID subspace, preserving the expressiveness of the original model. Concurrently, Kirichenko et al. demonstrated that models are able to learn both core features and spurious features. However, classifiers can rely upon spurious features, harming performance on minority groups. To reduce the reliance on spurious features, they propose to retrain the classifier on a small amount of "re-weighting" data, that allows the model to leverage the core features instead of the spurious features.

Other modifications and heuristics have also been proposed to improve fine-tuning, including side-tuning (Zhang et al., 2019), which tunes a small secondary network that is then combined with the original model, using larger/smaller learning rates for the classifier, as well as regularization-based methods (Jiang et al., 2020). We focus on the `LP+FT` protocol, as it is principled and achieves strong OOD performance.

Additionally, several works have studied properties of the model that influence the effectiveness of transfer learning (Azizpour et al., 2016; Huh et al., 2016; Kornblith et al., 2019; Lee et al., 2023a; Evci et al., 2022; Lee et al., 2023b; Izmailov et al., 2022; Lubana et al., 2023; Rame et al., 2022), including the robustness of pretrained features  (Salman et al., 2020; Utrera et al., 2021). While the connection between adversarial training and improved feature representations (Allen-Zhu & Li, 2021; Kaur et al., 2019) has been studied, we use virtual adversarial training during `LP` to learn a better classifier that is less reliant upon simple features, and we do not use an adversarially trained feature extractor. Finally, we note that while we are, to the best of our knowledge, the first to consider this holistic evaluation of safety and generalization in the context of transfer learning with highly expressive pretrained models, Hendrycks et al. have considered the trade-offs induced by different data augmentation strategies (Yun et al., 2019; Devries & Taylor, 2017; Hendrycks et al., 2020; Cubuk et al., 2019; 2020) on safety metrics in supervised learning. We emphasize that while our evaluation is similar, that our work focuses on a different context and contains an additional layer of complexity as we consider the interaction between adaptation protocols, generalization behavior and safety performance.

## B    EXPERIMENTAL DETAILS

Please see the https://github.com/pujacomputes/23-ICLR-Adaptation.git for training details. In brief, we performed grid-search to find the best parameters, which are as follows. For CIFAR-10 and CIFAR-100, we train only the classifier for 200 epochs with LR=30 during `LP`. For `FT`, the entire model is trained for 20 epochs with LR=1e-5. For `LP+FT`, the model's classifier is initialized with the solution found by `LP`, and then it is fine-tuned for 20 epochs. A grid-search was conducted to determine the LR for `LP` and `FT`. For Domain-Net Experiments, we use 200 epochs with LR=30 during `LP`. For `FT`, the entire model is trained for 20 epochs with LR=3e-4. For `LP+FT`, the model's classifier is initialized with the solution found by `LP`, and then it is fine-tuned for 20 epochs, using LR=3e-7. Furthermore, following Kumar et al., we freeze the batchnorm layers during `LP+FT`. A CLIP (Radford et al., 2021) pretrained ResNet-50 is used for the DomainNet experiments, while a MoCoV2 (He et al., 2020) is used for all CIFAR experiments. We use augmentation functions from timm (Wightman, 2019) and compute CKA scores using the packaged provided by torch-cka. When using augmented protocols, the same LRs are used. Note, all results were obtained by averaging over 3 seeds. We consider model soups of sizes 5,10,20, tune $\epsilon$ in 0.005, 0.01, 0.02 and 0.1 for UDP, and

$\alpha$ in 0.001, 0.01, 0.1 for VAT. For CIFAR-MNIST results, LP is done for 100 epochs, and FT is done for 20 epochs.

### B.1 MOTIVATION FOR HARDNESS-PROMOTING VARIANTS

We selected UDP (Pagliardini et al., 2022), VAT (Miyato et al., 2017), and model-soups (Wortsman et al., 2022) as simplicity bias mitigation strategies due to their effectiveness and ease of use. We emphasize, however, that our findings are not specific to the choice of a given mitigation strategy and we expect that advancements in such strategies will further improve the effectiveness of our proposed `LP+FT`variants. At present, the selected strategies are strong, representative mitigations that we have confirmed are effective at mitigating simplicity bias in the adaptation context using the synthetic dominoes dataset in Sec. 4.

We conceptually justify each strategy here:

- UDP is designed to help mitigate simplicity bias by learning by a large margin classifier, opposed to a narrow margin classifier that relies upon simple features. As noted by Shah et al. (2020), such narrow margin classifiers are sensitive to small perturbations and the simple features supporting the decision boundary may not be discriminative under distribution shifts. By maximizing uncertainty (instead of loss) to create adversarial perturbations, UDP is able to learn a maximum-margin classifier that is better able to handle such shifts. Notably, to create such a maximum-margin classifier, the model will necessarily learn more complex features;

- We use virtual adversarial training (VAT) to help avoid reliance upon simple features, as VAT enforces distribution smoothness so that classifiers become robust in some epsilon neighborhood around the input. We note that we are performing this training in the hidden representation space, so perturbations correspond may be altering high-level semantics. To maintain strong performance under such high-level perturbations, the model should learn to rely upon more complex features, and learn a better margin classifier;

- We use model-soups so that we may learn a set of classifiers that rely upon disjoint sets of features. By learning a set of diverse classifiers, we are able to average classifiers that have learned to rely upon different features, instead of becoming overly reliant upon a single simple feature. In future work, we intend to build a theoretical framework that helps us better justify these interventions and create new ones.

### B.2 APPLYING SIMPLICITY BIAS MITIGATION STRATEGIES TO FINE-TUNING STEP.

To demonstrate that simplicity bias mitigation strategies must be applied during the `LP` step of `FT` for maximum effectiveness, we conduct the following additional experiment.

*Setup.* We evaluate two additional protocols where `VAT` and `UDP` are applied only during the `FT` step, (`LP+FT(VAT)`, and `LP+FT(UDP)`), on the synthetic dominoes dataset. We plot the results for Randomized OOD Accuracy in Fig. 6.

*Results.* Here, we see that, across three different correlation ratios, `FT` variants lose performance with respect to the `LP` mitigation variants. Notably, `LP+ FT (UDP)` loses up to 4% performance with respect to `LP(UDP)+ FT`. While performance drops are not as large for `VAT`, we nonetheless see that `LP+ FT(VAT)` loses performance with respect to `LP(VAT)+ FT`.

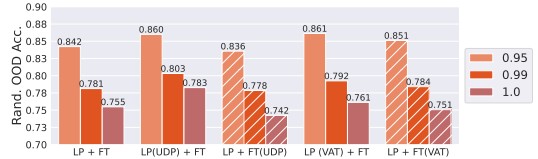

Figure 6: **Applying Mitigation Strategies to `FT`.** We create `FT` variants of our `LP` mitigation strategies and evaluate them on the synthetic dominoes dataset. We see that `FT` variants lose performance with respect to `LP` variants, indicating that interventions must be undertaken during the `LP` step as originally proposed.

Our results in Fig. 6 support our conceptual argument that mitigation strategies must be undertaken during the `LP` step to ensure that subsequent `FT` is in a direction that preserves complex features; applying mitigation strategies during `FT` may be too late to avoid simplicity bias. We note that

applying mitigation strategies during `FT`, in addition to `LP`, may further improve performance, and we will add these variants in the final version. We did not include a `FT` soup variant as it would be prohibitively expensive to train and average large soups of entire models (instead of classifiers). This highlights the computational efficiency of implementing mitigation strategies in the `LP` step itself.

## C ADDITIONAL RESULTS

Below, we include results corresponding to different hyperparameters (number of souped classifiers, $\alpha$ for vat, and $\delta$ for udp).

| Protocol | *Generalization* | | *Robustness* | | | *Calibration* | | | | *Anomaly Detection* | *Rep. Similarity* |
|---|---|---|---|---|---|---|---|---|---|---|---|
| | ID Acc. | OOD Acc. | C Acc. | $\overline{\text{C}}$ Acc. | Adv. Acc. | ID 1-RMS | C 1-RMS | $\overline{\text{C}}$ 1-RMS | OOD. 1-RMS | Out-of-Class AUROC | ID CKA |
| LP | 0.9138 | 0.8190 | 0.6912 | 0.6553 | 0.0003 | 0.9595 | 0.8303 | 0.8142 | 0.8696 | 0.6206 | 1.0000 |
| LP+ soup-5 | 0.9108 | 0.8348 | 0.7007 | 0.6678 | 0.0002 | 0.9748 | 0.8943 | 0.8835 | 0.9108 | 0.8463 | 1.0000 |
| LP+ soup-10 | 0.9129 | 0.8359 | 0.6985 | 0.6652 | 0.0003 | 0.9669 | 0.9104 | 0.8956 | 0.9205 | 0.8713 | 1.0000 |
| LP+ soup-20 | 0.9052 | 0.8353 | 0.6917 | 0.6588 | 0.0003 | 0.9605 | 0.9205 | 0.9037 | 0.9364 | 0.8859 | 1.0000 |
| LP+ udp-0.005 | 0.9129 | 0.8332 | 0.7015 | 0.6702 | 0.0003 | 0.9729 | 0.8879 | 0.8817 | 0.9017 | 0.8708 | 1.0000 |
| LP+ udp-0.01 | 0.9033 | 0.8356 | 0.6948 | 0.6643 | 0.0003 | 0.9689 | 0.9111 | 0.9023 | 0.9277 | 0.9033 | 1.0000 |
| LP+ udp-0.02 | 0.8885 | 0.8281 | 0.6796 | 0.6492 | 0.0004 | 0.9655 | 0.9259 | 0.9142 | 0.9473 | 0.9217 | 1.0000 |
| LP+ udp-0.1 | 0.8573 | 0.8005 | 0.6290 | 0.6064 | 0.0007 | 0.9245 | 0.9235 | 0.9143 | 0.9531 | 0.8570 | 1.0000 |
| LP+ vat-0.001 | 0.9189 | 0.8276 | 0.6945 | 0.6606 | 0.0006 | 0.9714 | 0.8564 | 0.8442 | 0.8927 | 0.7159 | 1.0000 |
| LP+ vat-0.01 | 0.8977 | 0.8251 | 0.6742 | 0.6483 | 0.0002 | 0.9265 | 0.9255 | 0.9139 | 0.9375 | 0.7200 | 1.0000 |
| FT | 0.9539 | 0.8754 | 0.7434 | 0.7553 | 0.0231 | 0.9668 | 0.8364 | 0.8453 | 0.9232 | 1.0000 | 0.6831 |
| LP+FT | 0.9442 | 0.8678 | 0.6921 | 0.6790 | 0.0018 | 0.9521 | 0.7849 | 0.7721 | 0.8864 | 0.6511 | 0.7853 |
| (LP+soup-5) +FT | 0.9466 | 0.8832 | 0.6997 | 0.6861 | 0.0001 | 0.9639 | 0.8197 | 0.8051 | 0.9155 | 0.9020 | 0.7603 |
| (LP+soup-10) +FT | 0.9467 | 0.8857 | 0.7022 | 0.6907 | 0.0001 | 0.9660 | 0.8307 | 0.8182 | 0.9184 | 0.9161 | 0.7671 |
| (LP+soup-20) +FT | 0.9466 | 0.8892 | 0.7031 | 0.6931 | 0.0001 | 0.9678 | 0.8390 | 0.8287 | 0.9216 | 0.9265 | 0.7806 |
| (LP+udp-0.005) +FT | 0.9458 | 0.8864 | 0.6962 | 0.6893 | 0.0005 | 0.9643 | 0.8127 | 0.8110 | 0.9119 | 0.9180 | 0.7742 |
| (LP+udp-0.01) +FT | 0.9450 | 0.8869 | 0.7048 | 0.6977 | 0.0004 | 0.9642 | 0.8335 | 0.8311 | 0.9209 | 0.9419 | 0.7746 |
| (LP+udp-0.02) +FT | 0.9440 | 0.8848 | 0.7028 | 0.6986 | 0.0004 | 0.9670 | 0.8472 | 0.8476 | 0.9237 | 0.9559 | 0.7764 |
| (LP+udp-0.1) + FT | 0.9435 | 0.8836 | 0.6959 | 0.6952 | 0.0000 | 0.9676 | 0.8449 | 0.8525 | 0.9355 | 0.9651 | 0.7382 |
| (LP+vat)+FT | 0.9611 | 0.8900 | 0.7442 | 0.7321 | 0.0027 | 0.9294 | 0.8355 | 0.8281 | 0.9178 | 0.8276 | 0.7839 |

Table 6: **CIFAR10, Hardness-Promoting Augmentations.**

| Protocol | *Generalization* | | *Robustness* | | | *Calibration* | | | | *Anomaly Detection* | *Rep. Similarity* |
|---|---|---|---|---|---|---|---|---|---|---|---|
| | ID Acc. | OOD Acc. | C Acc. | $\overline{\text{C}}$ Acc. | Adv. Acc. | ID 1-RMS | C 1-RMS | $\overline{\text{C}}$ 1-RMS | OOD. 1-RMS | Out-of-Class AUROC | ID CKA |
| LP | 0.9521 | 0.8124 | 0.7010 | 0.7378 | 0.2350 | 0.9313 | 0.8693 | 0.8802 | 0.9117 | 0.9907 | 1.0000 |
| LP+ udp-0.005 | 0.9524 | 0.8114 | 0.7012 | 0.7379 | 0.2337 | 0.9304 | 0.8699 | 0.8806 | 0.9108 | 0.9907 | 1.000 |
| LP+ udp-0.01 | 0.9524 | 0.8110 | 0.7017 | 0.7382 | 0.2353 | 0.9308 | 0.8691 | 0.8801 | 0.9118 | 0.9908 | 1.000 |
| LP+ udp-0.02 | 0.9500 | 0.8126 | 0.7036 | 0.7387 | 0.2373 | 0.9343 | 0.8621 | 0.8763 | 0.9135 | 0.9913 | 1.000 |
| LP+ udp-0.1 | 0.9459 | 0.8165 | 0.6840 | 0.7220 | 0.2339 | 0.9032 | 0.8243 | 0.8427 | 0.8990 | 0.9882 | 1.000 |
| LP+ soup-5 | 0.9439 | 0.7996 | 0.6874 | 0.7290 | 0.2451 | 0.8806 | 0.7868 | 0.8094 | 0.9064 | 0.9897 | 1.0000 |
| LP+ soup-10 | 0.9373 | 0.7904 | 0.6767 | 0.7220 | 0.2547 | 0.8496 | 0.7478 | 0.7709 | 0.8841 | 0.9887 | 1.0000 |
| LP+ soup-20 | 0.9298 | 0.7841 | 0.6601 | 0.7082 | 0.2575 | 0.8056 | 0.7084 | 0.7305 | 0.8274 | 0.9867 | 1.0000 |
| LP+ vat-0.001 | 0.9524 | 0.8122 | 0.7010 | 0.7379 | 0.2345 | 0.9299 | 0.8682 | 0.8791 | 0.9103 | 0.9907 | 1.0000 |
| FT | 0.9518 | 0.7168 | 0.7011 | 0.7164 | 0.1563 | 0.8873 | 0.9019 | 0.8604 | 0.9295 | 0.9794 | 0.7847 |
| LP+FT | 0.9643 | 0.8261 | 0.7426 | 0.7671 | 0.2135 | 0.9782 | 0.9472 | 0.9451 | 0.8742 | 0.9924 | 0.9887 |
| (LP+udp-0.005) +FT | 0.9627 | 0.8243 | 0.7434 | 0.7666 | 0.2153 | 0.9811 | 0.9456 | 0.9445 | 0.8736 | 0.9922 | 0.98950 |
| (LP+udp-0.01) +FT | 0.9627 | 0.8253 | 0.7436 | 0.7669 | 0.2133 | 0.9812 | 0.9454 | 0.9447 | 0.8737 | 0.9923 | 0.98957 |
| (LP+udp-0.02) +FT | 0.9637 | 0.8265 | 0.7448 | 0.7681 | 0.2157 | 0.9768 | 0.9464 | 0.9467 | 0.8757 | 0.9927 | 0.98927 |
| (LP+udp-0.1) +FT | 0.9614 | 0.8249 | 0.7499 | 0.7689 | 0.2165 | 0.9808 | 0.9441 | 0.9420 | 0.8711 | 0.9912 | 0.9861 |
| (LP+soup-5) + FT | 0.9608 | 0.8163 | 0.7456 | 0.7684 | 0.1855 | 0.9760 | 0.9498 | 0.9492 | 0.8678 | 0.9936 | 0.98540 |
| (LP+soup-10) + FT | 0.9580 | 0.8114 | 0.7445 | 0.7678 | 0.1753 | 0.9838 | 0.9503 | 0.9488 | 0.8748 | 0.9938 | 0.98360 |
| (LP+soup-20) + FT | 0.9594 | 0.8165 | 0.7450 | 0.7684 | 0.1782 | 0.9893 | 0.9503 | 0.9490 | 0.8609 | 0.9936 | 0.98190 |
| (LP+vat-0.001) +FT | 0.9647 | 0.8247 | 0.7425 | 0.7650 | 0.2224 | 0.9727 | 0.9521 | 0.9463 | 0.8775 | 0.9925 | 0.9370 |

Table 7: **Living17, Hardness-Promoting Augmentations**

| Protocol | *Generalization* | | *Robustness* | | | *Calibration* | | | | *Anomaly Detection* | *Rep. Similarity* |
|---|---|---|---|---|---|---|---|---|---|---|---|
| | ID Acc. | OOD Acc. | Sketch-C Acc. | Real-C Acc. | Adv. Acc. | ID 1-RMS | Sketch-C 1-RMS | Real-C 1-RMS | OOD. 1-RMS | Out-of-Class AUROC | ID CKA |
| LP | 0.8913 | 0.8013 | 0.6019 | 0.6020 | 0.1768 | 0.9638 | 0.9264 | 0.9045 | 0.9014 | 0.8679 | 1.0000 |
| LP+augmix | 0.8897 | 0.7998 | 0.6336 | 0.6104 | 0.1872 | 0.9718 | 0.9230 | 0.9263 | 0.9083 | 0.8818 | 1.0000 |
| LP+autoaug | 0.8944 | 0.8057 | 0.6419 | 0.6257 | 0.1857 | 0.9614 | 0.9357 | 0.9309 | 0.9022 | 0.8849 | 1.0000 |
| LP+randaug | 0.8971 | 0.8090 | 0.6392 | 0.6232 | 0.1877 | 0.9559 | 0.9321 | 0.9312 | 0.9036 | 0.8875 | 1.0000 |
| LP+vat | 0.8836 | 0.7914 | 0.5893 | 0.5963 | 0.1687 | 0.8897 | 0.9552 | 0.8905 | 0.9178 | 0.8735 | 1.0000 |
| FT | 0.7613 | 0.4522 | 0.5186 | 0.2744 | 0.4164 | 0.8368 | 0.6379 | 0.7234 | 0.5597 | 0.8841 | 0.6092 |
| FT+augmix | 0.8246 | 0.5233 | 0.5911 | 0.3408 | 0.4802 | 0.9308 | 0.8042 | 0.8665 | 0.6761 | 0.9255 | 0.5272 |
| FT+autoaug | 0.7786 | 0.5161 | 0.5561 | 0.3160 | 0.4313 | 0.9157 | 0.7485 | 0.8246 | 0.7324 | 0.9231 | 0.7025 |
| FT+randaug | 0.7823 | 0.5370 | 0.5610 | 0.3298 | 0.4551 | 0.9160 | 0.7970 | 0.8682 | 0.7444 | 0.9318 | 0.6477 |
| LP+FT | 0.8985 | 0.7990 | 0.6343 | 0.5979 | 0.1927 | 0.9566 | 0.8899 | 0.8445 | 0.8024 | 0.9022 | 0.9222 |
| LP+(FT+augmix) | 0.9047 | 0.8081 | 0.6673 | 0.5980 | 0.2597 | 0.9768 | 0.9200 | 0.9067 | 0.8443 | 0.9155 | 0.8811 |
| LP+(FT+autoaug) | 0.9023 | 0.8028 | 0.6571 | 0.5851 | 0.2354 | 0.9830 | 0.9249 | 0.8990 | 0.8484 | 0.9034 | 0.9096 |
| LP+(FT+randaug) | 0.9054 | 0.8099 | 0.6703 | 0.6152 | 0.2489 | 0.9786 | 0.9194 | 0.9044 | 0.8598 | 0.9252 | 0.9000 |
| (LP+vat) +FT | 0.9048 | 0.8009 | 0.6466 | 0.6131 | 0.1942 | 0.9686 | 0.8911 | 0.8428 | 0.7985 | 0.9204 | 0.9370 |
| (LP+vat) +(FT+augmix) | 0.9032 | 0.8024 | 0.6589 | 0.5896 | 0.2525 | 0.9769 | 0.9169 | 0.8929 | 0.8384 | 0.9212 | 0.8673 |
| (LP+vat)+(FT+autoaug) | 0.9003 | 0.8049 | 0.6600 | 0.5862 | 0.2331 | 0.9783 | 0.9178 | 0.9000 | 0.8381 | 0.9149 | 0.9244 |
| (LP+vat)+(FT+randaug) | 0.9006 | 0.8060 | 0.6651 | 0.5894 | 0.2622 | 0.9762 | 0.9197 | 0.8993 | 0.8414 | 0.9238 | 0.8956 |

Table 8: **DomainNet, Diversity Promoting Augmentations and Generalization Trade-offs.**

| Protocol | *Generalization* | | *Robustness* | | | *Calibration* | | | | *Anomaly Det.* | *Rep. Similarity* |
|---|---|---|---|---|---|---|---|---|---|---|---|
| | ID Acc. | OOD Acc. | C Acc. | $\overline{\text{C}}$ Acc. | Adv. Acc. | ID 1-RMS | C 1-RMS | $\overline{\text{C}}$ 1-RMS | OOD. 1-RMS | Out-of-Class AUROC | ID CKA |
| LP | 0.9297 | 0.9083 | 0.8532 | 0.7491 | 0.7077 | 0.9794 | 0.9006 | 0.9007 | 0.9301 | 0.9623 | 0.0668 |
| LP+ soup-5 | 0.9220 | 0.9151 | 0.8315 | 0.7432 | 0.7050 | 0.9598 | 0.9232 | 0.9279 | 0.9623 | 0.9665 | 0.1399 |
| LP+ soup-10 | 0.9156 | 0.9135 | 0.8183 | 0.7344 | 0.6985 | 0.9476 | 0.9221 | 0.9271 | 0.9732 | 0.9602 | 0.1778 |
| LP+ soup-20 | 0.9069 | 0.9064 | 0.8065 | 0.7216 | 0.6885 | 0.9279 | 0.9129 | 0.9191 | 0.9714 | 0.9484 | 0.2617 |
| LP+ udp-0.005 | 0.9299 | 0.9092 | 0.8533 | 0.7494 | 0.7079 | 0.9794 | 0.9009 | 0.9003 | 0.9312 | 0.9614 | 0.0822 |
| LP+ udp-0.01 | 0.9298 | 0.9097 | 0.8535 | 0.7495 | 0.7083 | 0.9795 | 0.9007 | 0.9006 | 0.9316 | 0.9616 | 0.0880 |
| LP+ udp-0.02 | 0.9294 | 0.9108 | 0.8538 | 0.7497 | 0.7088 | 0.9789 | 0.9012 | 0.9014 | 0.9335 | 0.9631 | 0.1017 |
| LP+ udp-0.1 | 0.9238 | 0.9218 | 0.8377 | 0.7488 | 0.7111 | 0.9801 | 0.9154 | 0.9216 | 0.9517 | 0.9645 | 0.1478 |
| LP+ vat-0.001 | 0.9298 | 0.9091 | 0.8533 | 0.7493 | 0.7078 | 0.9801 | 0.9014 | 0.9012 | 0.9325 | 0.9614 | 0.0784 |
| LP+ vat-0.01 | 0.9295 | 0.9094 | 0.8531 | 0.7494 | 0.7080 | 0.9800 | 0.9039 | 0.9040 | 0.9342 | 0.9632 | 0.0837 |
| LP+ vat-0.1 | 0.9275 | 0.9106 | 0.8493 | 0.7481 | 0.7087 | 0.9581 | 0.9191 | 0.9246 | 0.9589 | 0.9598 | 0.1528 |
| FT | 0.9724 | 0.8761 | 0.9218 | 0.8131 | 0.8074 | 0.9577 | 0.8429 | 0.8418 | 0.8855 | 0.9138 | 0.9317 |
| LP+FT | 0.9692 | 0.9387 | 0.9195 | 0.8106 | 0.7736 | 0.9451 | 0.8034 | 0.7743 | 0.9026 | 0.8949 | 0.5349 |
| (LP+soup-5) +FT | 0.9685 | 0.9417 | 0.9210 | 0.8136 | 0.7787 | 0.9385 | 0.8079 | 0.7765 | 0.9102 | 0.8974 | 0.5315 |
| (LP+soup-10) +FT | 0.9681 | 0.9411 | 0.9220 | 0.8178 | 0.7824 | 0.9382 | 0.8119 | 0.7796 | 0.9072 | 0.8933 | 0.5521 |
| (LP+soup-20) +FT | 0.9677 | 0.9395 | 0.9213 | 0.8164 | 0.7837 | 0.9385 | 0.8107 | 0.7817 | 0.9070 | 0.8964 | 0.5411 |
| (LP+udp-0.005) +FT | 0.9677 | 0.9297 | 0.9142 | 0.8104 | 0.7710 | 0.9422 | 0.8024 | 0.7718 | 0.8942 | 0.8916 | 0.6428 |
| (LP+udp-0.01) +FT | 0.9677 | 0.9359 | 0.9195 | 0.8098 | 0.7721 | 0.9417 | 0.8029 | 0.7732 | 0.9019 | 0.8999 | 0.4239 |
| (LP+udp-0.02) +FT | 0.9687 | 0.9349 | 0.9195 | 0.8136 | 0.7724 | 0.9437 | 0.8067 | 0.7736 | 0.8994 | 0.8981 | 0.5015 |
| (LP+udp-0.1) +FT | 0.9688 | 0.9423 | 0.9242 | 0.8174 | 0.7811 | 0.9408 | 0.8130 | 0.7815 | 0.9072 | 0.9064 | 0.4496 |
| (LP+vat-0.001)+FT | 0.9681 | 0.9366 | 0.9180 | 0.8111 | 0.7727 | 0.9422 | 0.8033 | 0.7732 | 0.9013 | 0.8962 | 0.5904 |
| (LP+vat-0.01)+FT | 0.9689 | 0.9366 | 0.9168 | 0.8121 | 0.7766 | 0.9455 | 0.8062 | 0.7791 | 0.9013 | 0.8918 | 0.5687 |
| (LP+vat-0.1)+FT | 0.9692 | 0.9402 | 0.9207 | 0.8127 | 0.7743 | 0.9420 | 0.8068 | 0.7734 | 0.9083 | 0.8978 | 0.4398 |

Table 9: **CIFAR10 with Resnet101/SimCLR Pretrained Model.** We see that with a larger model, and different pretraining method, our proposed variants still have some benefits. We note that the baseline performance is also improved as a result of a more larger pretrained model.

