# OpenReview forum: "A Closer Look at Model Adaptation using Feature Distortion and Simplicity Bias"
_ICLR.cc/2023/Conference — ICLR 2023 notable top 25%_

### Official Review · Reviewer_79RY · 2022-10-23

**Confidence:** 3
**Correctness:** 4
**Technical Novelty And Significance:** 3
**Empirical Novelty And Significance:** 3
**Recommendation:** 8

**Clarity, Quality, Novelty And Reproducibility:**

The clarity of the paper is good, however the paper lacks in novelty in terms of techniques proposed. Though the empirical analysis is strong and can be a good addition in understanding OOD generalization.

**Strength And Weaknesses:**

Strengths:

	- The paper looks at additional metrics than solely the OOD generalization accuracy which is critical for model deployment. Although LP + FT  leads to less feature distortion and is a sweet spot for strong ID and OOD accuracy, it is not the best method for achieving good performance on the other metrics such as calibration / robustness.

	- The authors conduct several experiments to understand the interplay between feature distortion and simplicity bias. In principle, the authors find that under slightly less correlation, a small amount of feature distortion is good for mitigating simplicity bias. There are similar empirical observations throughout the paper which are informative.

	- The proposed plugin modules for LP leads to good improvements over FT or LP+FT, for Rand. OOD accuracy, which shows that these strategies can be leveraged to mitigate simplicity bias. Also these modules lead to improvements in the other metrics such as robustness and calibration.

While the paper does an extensive study on feature distortion and simplicity bias, I have certain comments on the weaknesses/comments./questions of the paper:

Weakness:
	- The proposed method for improving LP or LP+FT,  though motivated are not completely new, but are already existing in the literature.

	- It would be beneficial if the authors can give more intuition on the specific choice of pre-trained models? The paper would be strengthened if a larger variety of pre-trained models are evaluated on.

	- Would the proposed strategies also help in improving performance on OOD derivatives of the Imagenet validation set (e.g., Imagenet-A, Imagenet-C) . For e.g., train with LP+FT using the strategies on the training set of Imagenet and compute downstream test performance on Imagenet-A, C,R and ObjectNet?


**Summary Of The Paper:**

The paper empirically studies model adaptation with additional metrics such as calibration, robustness in addition to ID/OOD generalization. The study is conducted through the lens of simplicity bias and feature distortion, via studying the characteristics of linear probing and fine-tuning. Finally, the authors propose strategies to use during linear-probing which can decrease the simplicity bias and improve on OOD generalization and the additional metrics studied in the paper.

**Summary Of The Review:**

I think that overall, the paper is a strong empirical paper, inspite of the weaknesses stated. The insights from the paper that mitigating feature distortion might not be enough for metrics such as calibration etc and there exists a need to reduce the simplicity bias, are important.

---

> ### Author Response · Authors · 2022-11-14
> **Response to Reviewer 79RY**
>
> Thank you for the helpful feedback and questions! We are encouraged that the insights from the paper were found to be “important” and the empirical analysis considered “strong.” Below, we address all questions and concerns.
>
> >The proposed methods for improving LP or LP+FT, though motivated, are not completely new, but are already existing in the literature.
>
> Thanks for the comment! We use these strategies to support our claim that protocols which help control simplicity bias induce safer models. While we selected these particular strategies (UDP, VAT, Soups) for their effectiveness and ease of use, ***our findings are not specific to a given strategy*** and we expect that ***advancements in such strategies will further improve*** the effectiveness of our proposed LP+FT variants.
>
> We also emphasize that our primary contribution is not the proposal of any particular LP+FT variant, but ***the creation of new insights for the design of novel adaptation protocols.*** In particular, our expanded evaluation, which identified shortcomings in existing protocols as well as the feature distortion perspective, will be valuable for studying the efficacy of new protocols. Moreover, our investigation into the role of simplicity bias in inducing safe adaptation can help inform the design of such novel protocols.
>
> >It would be beneficial if the authors can give more intuition on the specific choice of pre-trained models? The paper would be strengthened if a larger variety of pre-trained models are evaluated on.
>
> Thanks for the question! We use ResNet50s pretrained with MoCo-V2 and CLIP for our experiments as this architecture/pretraining combination is known to provide high-quality representations that are capable of strong ID and OOD performance when performing transfer learning [1]. We note that *high quality representations* are required as low quality representations are not capable of spanning both the in and out distribution feature subspaces. In such situations, LP+FT will struggle to find a solution that has superior out-of-distribution performance than LP.
>
> **Additional Experimental Verification:** To further demonstrate that our results are not specific to the choice of architecture or pre-training strategy, we have added *additional results for a ResNet-101 pre-trained using SimCLR evaluated on CIFAR10* to the revised manuscript (Table 9, page 16). Using a different high-quality model, we find that our proposed LP+FT variants *continue to bring some benefits* to safety and generalization performance.
>
> **Claims are Agnostic to Architecture/Pretraining Choice:** While we will add ViT experiments to the final version, we strongly emphasize that our central claims are agnostic to the choice of underlying architecture and pre-training paradigm. *Indeed, our central claim, that simplicity bias and feature distortion must be mitigated to induce safe and strong adaptation, is independent of these factors as it is the adaptation protocol that is responsible for inducing susceptibility to these pitfalls on downstream dataset.* For example, without intervention, SGD is known to prefer simple, linearly separable features. This implies that the fine-tuning protocol will remain susceptible to inducing simplicity bias, even if the pretrained model has been improved! While we acknowledge that better models may be less inherently predisposed to such pitfalls on standard benchmarks, our findings remain important as we do not know apriori how susceptible a particular target dataset is to these pitfalls.
>
> [1] Fine-Tuning can Distort Pretrained Features and Underperform Out-of-Distribution. Ananya Kumar, Aditi Raghunathan, Robbie Jones, Tengyu Ma, Percy Liang. In ICLR 2022.
>
> >Would the proposed strategies also help in improving performance on OOD derivatives of the Imagenet validation set (e.g., Imagenet-A, Imagenet-C) . For e.g., train with LP+FT using the strategies on the training set of Imagenet and compute downstream test performance on Imagenet-A, C,R and ObjectNet?
>
> We expect that our proposed variants will provide some additional benefits over vanilla LP+FT on these ImageNet derivatives. Indeed, the vanilla LP+FT protocol does improve performance on these OOD derivatives [1], so *we expect that our variants will further improve upon this performance.* We continue to expand the manuscript with additional architectures and datasets for the final version, including the suggested derivatives as well as ResNet-101 and ViT experiments!
>
> [1] Fine-Tuning can Distort Pretrained Features and Underperform Out-of-Distribution. Ananya Kumar, Aditi Raghunathan, Robbie Jones, Tengyu Ma, Percy Liang. In ICLR 2022.

---

> > ### Comment · Reviewer_79RY · 2022-11-16
> > **Response to Authors**
> >
> > I thank the authors for their detailed reply. I would like to maintain my score and vouch for its acceptance, considering the thorough empirical analysis done by the paper.

---

### Official Review · Reviewer_QKt7 · 2022-10-24

**Confidence:** 3
**Correctness:** 3
**Technical Novelty And Significance:** 3
**Empirical Novelty And Significance:** 4
**Recommendation:** 10

**Clarity, Quality, Novelty And Reproducibility:**

Clarity:
The paper was moderately clear, but there is significant room for improvement.  The discussion of the pros and cons of FT vs. LP are buried in the text, and not as easy to extract as they could be; It might help to outline the structure of the paper including the core claims (some of which would be novel and some of which were previously demonstrated), along the lines of the summary I provided.  Feature distortion could be explained more in the background section.  It wasn’t entirely clear why the “hardness promoting” methods are aiming to accomplish or why they will accomplish it.  There are a few typos I stumbled on.  What do bold/underline represent in the tables?

Quality:
The paper is high quality.  The experiments are informative and thorough, including synthetic and real datasets.  They effectively support the central claims, which are noteworthy.

Novelty:
The novelty is moderate.  The paper is an in-depth analysis of how simple model adaptation approaches work and how they might fail.  Nonetheless, this is a significant contribution given recent results suggesting these simple model adaptation approaches are highly effective.
The weaknesses of these approaches identified in this work are important for the community to know about.


**Strength And Weaknesses:**


Strengths:
- The topic is significant and timely.  Robustness is a central concern in deep learning, and OOD robustness is a hot topic.  This paper identifies safety limitations of LP/FT/LP+FT that could be easy to overlook, given the strong performance of these methods (and especially LP+FT) on OOD generalization.
- The investigation is insightful and effectively supports the paper’s claims.  Besides identifying these weaknesses, the paper provides a convincing explanation for them.


Weaknesses:
- The improvements in Section 5 aren’t that big.
- The proposed mitigations aren’t clearly justified.

**Summary Of The Paper:**

This paper studies model adaptation.
It identifies simplicity bias as a weakness of the "linear probe" (LP) approach, which learns a linear classifier on top of pre-trained features.
Feature distortion was previously identified as a weakness of the alternative approach of fine-tuning all of the pretrained weights (FT).
This led previous work to propose doing LP followed by fine-tuning the other weights (LP+FT).
This work shows that, while LP+FT addresses the issue of feature distortions present in FT, it may fail to address the issue of simplicity bias present in LP.

This observation is supported by experiments on synthetic and real data.
And this leads the authors to propose methods of “hardness promoting” changes to the LP step of LP+FT.
Further experiments on synthetic and real data demonstrate the effectiveness of these approaches.
Although the improvements on real data are perhaps relatively minor, they help demonstrate the scientific claim of the paper.

**Summary Of The Review:**

The paper makes a solid and significant contribution to our understanding of model adaptation, which is central to modern large-scale deep learning.  It highlights, explains, and takes some small steps towards mitigating safety issues with leading approaches to model adaptation.

---

> ### Author Response · Authors · 2022-11-14
> **Response to Reviewer QKt7 (Part 1)**
>
> Thank you for your insightful comments! We are delighted that you found our paper to be of “high quality” with “thorough and informative” experiments that “effectively support” our “noteworthy” claims. Below, we address all concerns and questions.
>
> ***Response to Weakness:***
> >“The improvements in Section 5 aren’t that big.”
>
> Thanks for the comment! On real datasets, it is more challenging to distinguish between simple and complex features, which can make it difficult to see large benefits since the vanilla solution is sometimes well-suited for these datasets. *Nonetheless, our central claim, that both simplicity bias and feature distortion should be mitigated for strong and sample generalization, remains important as we cannot know **a priori** if vanilla solution will be sufficient for **arbitrary** datasets.* Our experiments with the synthetic dominoes datasets demonstrate this to a large extent. In this sense, *our variants **preemptively protect** against this pitfall.* Moreover, while we selected UDP/VAT/model-soups for their effectiveness and ease-of-use, the design of more effective mitigation strategies remains an active direction of research. As the quality of such mitigation approaches improves, *we fully expect that implementing the LP step with these improved strategies will lead to larger and more consistent improvements.* We’ve added this discussion to Sec. 5 (pg. 9) and intend to keep updating this manuscript (and the codebase) in the future with improved mitigation strategies.
>
> >- “The proposed mitigations aren’t clearly justified."
> >- "It wasn’t entirely clear why the “hardness promoting” methods are aiming to accomplish or why they will accomplish it.”
>
> Thanks for the comment! We address both these questions about the mitigation strategies together. We selected these particular mitigations (UDP, VAT, Soups) for their *effectiveness and ease of use.* We emphasize, however, that *our findings are **not specific** to a given mitigation strategy* and we expect that advancements in such strategies will further improve the effectiveness of our proposed LP+FT variants. At present, the selected strategies are strong, representative mitigations that we have confirmed are effective at mitigating simplicity bias in the adaptation context using the synthetic experiment.
>
> **Motivation for UDP:** UDP is designed to help mitigate simplicity bias by learning by a large margin classifier, opposed to a narrow margin classifier that relies upon simple features. As noted by [1], such narrow margin classifiers are sensitive to small perturbations and the simple features supporting the decision boundary may not be discriminative under distribution shifts. By maximizing uncertainty (instead of loss) to create adversarial perturbations, *UDP is able to learn a maximum-margin classifier that is better able to handle such shifts.* Notably, to create such a maximum-margin classifier, the *model will necessarily learn more complex features.*
>
> **Motivation for VAT:** We use virtual adversarial training to help avoid reliance upon simple features, as *VAT enforces distribution smoothness* so that classifiers become robust in some epsilon neighborhood around the input. We note that we are performing this training in the hidden representation space, so perturbations correspond *may be altering high-level semantics.* To maintain strong performance under such high-level perturbations, the model should learn to rely upon more complex features, and learn a better margin classifier.
>
> **Motivation for Soup:** We use model-soups so that we may learn a *set of classifiers that rely upon disjoint sets of features.* By first learning a set of diverse classifiers, and then averaging over them, our expectation is that the average classifier will not be overly reliant upon a single, overly simple feature. Instead, we expect that the average classifier will rely upon several different semantically-high level features, leading to a better supported decision boundary.
>
> While we have included conceptual justification and empirical justification for why these strategies were selected, we intend to build a *theoretical framework* that helps us better justify these interventions and create new ones in future work. **We’ve also added this discussion to the manuscript (Supp. C).**
>
> [1] The Pitfalls of Simplicity Bias in Neural Networks. Harshay Shah, Kaustav Tamuly, Aditi Raghunathan, Prateek Jain, Praneeth Netrapalli. In NeurIPS 2020.
>
> ***Summary:*** While we selected three representative mitigation strategies for their ease-of-use and effectiveness, our central claim is agnostic to the particular choice of mitigation strategy and we leave a theoretical justification of their effectiveness to future work. We further suspect that the performance of our proposed LP+FT variants will improve as more sophisticated mitigation strategies are proposed.

---

> > ### Author Response · Authors · 2022-11-14
> > **Response to Reviewer QKt7 (Part 2)**
> >
> > ***Response to Concerns about Clarity/Quality/Novelty/Reproducibility***
> >
> > >“The paper was moderately clear, but there is significant room for improvement. The discussion of the pros and cons of FT vs. LP are buried in the text, and not as easy to extract as they could be; Feature distortion could be explained more in the background section.”
> >
> > Thanks for the suggestion! We have updated the paper so that each ***point is explained in a dedicated sub-section*** (Obs 1: Mitigating Feature Distortion does not induce safe adaptation, and Obs 2: Linear Probing Solutions Matter) with an ***accompanying explanatory figure*** (Fig.2: Disparate Performance of Adaptation Protocols and Fig. 3: Dataset-Level Distortion). Moreover, we have moved the discussion of simplicity bias to the next section to improve the clarity of Section 3.
> >
> > >"What do bold/underline represent in the tables?"
> >
> > Thanks for the catch! Bold represents best performance, while underline represents the second best performance. We’ve added the explanations to the captions as well.

---

### Official Review · Reviewer_AzrZ · 2022-10-25

**Confidence:** 4
**Correctness:** 4
**Technical Novelty And Significance:** 3
**Empirical Novelty And Significance:** 4
**Recommendation:** 8

**Clarity, Quality, Novelty And Reproducibility:**

- From the manuscript, it is not very clear to me what motivated connecting the empirical findings about the disparities in the safety metrics with the notion of simplicity bias. Moreover, would the same findings hold in case different metrics and / or datasets / architectures were considered?
- As mentioned in the previous section, I have concerns regarding the experimental setting currently adopted in the work as I found it too limited and not supporting some claims in the manuscript. It is important to emphasize here that I do not think performing experiments with the setting considered in this work in a weakness or issue per se, my concern is that the contributions are solely based on the experiments, which have a narrow scope, restricting the strength and generalization of the findings. Also, the authors claim to be studying large-scale settings, which does not seem to be the case to me.
- Even though Table 3 indicates the strategies to mitigate simplicity bias were helpful to improve the safety metrics, it is not clear whether those improvements are in fact a consequence of attenuating simplicity bias in the studies cases.
  - Perhaps one way to show this empirically could be reporting the gap in classification performance between the hardest and easiest classes (in terms of accuracy) in each dataset for models trained with and without the bias mitigation strategies. I believe this could be seen as a proxy measure for how vulnerability to the simplicity bias issue changed in each case.
- The manuscript also lacks clarity due to an excess of acronyms (some of them were not defined in the text prior to use, such as DNNs), tables font is too small, and the text presents several typos. For example:
  - Page 2, figure 1: Anamolies -> Anomalies
  - Page 4: comprising -> compromising?
  - Page 5: sensitivity -> sensitive?
  - Add spaces between acronyms and the following word: e.g. Page 2: FPand LP -> FP and LP


**Strength And Weaknesses:**

- Strength
  - The work tackles a relevant problem for the community: how to best leverage large pre-trained models to downstream tasks where safety-related metrics are also important?
  - The empirical analysis is extensive in the sense it considers three protocols for employing pre-trained models, across three datasets, and multiple metrics. The authors explained those findings through the lens of simplicity bias in neural networks.

- Weaknesses
  - The contributions of this work are mostly empirical, which *is not a weakness per se*. My concern regarding this aspect is that the experiments were performed considering a rather restrictive, and perhaps also outdated, setting, which makes the findings of the work less relevant for the community. More specifically, the only architecture used throughout the experiment was a ResNet-50 trained with MoCo-V2 / CLIP using ImageNet-1k. Besides being very limited from an experimental perspective, this setting also does not match current trends in the community. I believe it is to diversify the types of architectures and pre-training tasks in order to increase the strength of the conclusions in this work. For example, I suggest the authors include experiments with ResNets-101 / 152 and Vision transformers, pre-trained with tasks such as SimCLR, and datasets such as ImageNet-21k.

  - The above mentioned concern gets increased relevance given that in several parts of the manuscript the authors claim to be studying large-scale models. For example, in page 8, the authors mentioned "In this work, we take a closer look at the behavior of protocols designed for adapting large-scale pretrained models to downstream datasets" ), however, given that currently the term large-scale suggests bigger models and datasets, I am not confident I can agree the studied setting can be deemed *large-scale*.

  - The manuscript lacks clarity mostly due to an excess of acronyms and typos (see next section for details).

**Summary Of The Paper:**

In this work the authors investigate protocols for utilizing pre-trained models for downstream tasks. Motivated by the observation that different protocols perform differently according to several metrics under varying levels of distribution shift, the authors propose to analyse such protocols under the light of simplicity bias, i.e. they verify that certain protocols tend yield simpler features. In order to improve the use of pre-trained models in terms of the considered safety metrics, the authors then propose to leverage training with perturbed versions of the input data, as well as a combination of multiple linear probes by averaging their weights (models soup). Experiments showed, for example, that in-distribution performance and safety in terms of the reported metrics are not consistent across protocols.


**Summary Of The Review:**

This work investigates out-of-distribution generalization and safety aspects of different protocols for applying pre-trained models to downstream tasks. The authors empirically found that different approaches yield models with different performance in terms of out-of-distribution generalization and safety in terms of selected metrics. Those findings are explained through the lens of simplicity bias and I found this connection insightful, albeit a little not very well-motivated. My major concern with this work are its limitations with respect to the experimental protocol, especially because all conclusions are based solely on empirical evidence. Unfortunately, the authors only considered one type of architecture, two self-supervised tasks, and the ImageNet-1k dataset for pre-training. I believe this renders the findings in this submission limited and not sound. Moreover, the current manuscript contains presentations issues that compromise the clarity of the work. All in all, I believe the weaknesses of this work currently outweigh the merits, and I thus believe it is marginally below the acceptance bar.

---

> ### Author Response · Authors · 2022-11-14
> **Response to Reviewer AzrZ (Part 1)**
>
> We thank the reviewer for their detailed comments! We are encouraged that they found our work to tackle “a relevant problem for the community” and the lens of simplicity bias “insightful” when studying this problem. Below, we address all questions and concerns.
>
> ***Responses to Weaknesses:***
>
> >"My concern regarding this aspect is that the experiments were performed considering a rather restrictive, and perhaps also outdated, setting, which makes the findings of the work less relevant for the community."
>
> Thanks for the comment! We have *added results for an additional architecture **(ResNet-101) and pretraining strategy (SimCLR)** on CIFAR10 to the revised manuscript (Table 9, page 16).* For the final version, we will also add results for ViTs.
>
> We emphasize, however, that ResNet50 remains a work-horse architecture for many applications and a model-of-choice for many practitioners. Moreover, in resource constrained scenarios, ResNet50s can outperform ViTs as the latter may be challenging to train due to ***large memory requirements.*** We further note that the two pretraining strategies used in our paper, MoCoV2 (March 2020) and CLIP (Jan. 2021), were both ***released after*** the recommended additional strategy, SimCLR (Feb. 2020), and ***surpass*** its performance.
>
> > “More specifically, the only architecture used throughout the experiment was a ResNet-50 trained with MoCo-V2 / CLIP using ImageNet-1k.”
>
> We note that, while MoCo-V2 was trained on ImageNet-1K, ***CLIP was trained using 400 million (image, text) pairs*** harvested from the internet.
>
> > - I believe it is to diversify the types of architectures and pre-training tasks in order to increase the strength of the conclusions in this work. For example, I suggest the authors include experiments with ResNets-101 / 152 and Vision transformers, pre-trained with tasks such as SimCLR, and datasets such as ImageNet-21k.
>
> Thank you for the comments! We provide additional empirical verification and conceptual justification to address these comments below.
>
> **Additional Experimental Verification:** To further demonstrate that our claims hold for bigger models and different pre-training paradigms, we use (as per the reviewer’s suggestion) a ***ResNet-101 trained with SimCLR** as the underlying pretrained model.* When this larger model is adapted to CIFAR10 (Table 9, Page 16), we see that our *proposed variants continue to bring some benefits* to both safety and generalization performance. We will add Living17 and DomainNet experiments for ResNet-101 to the final version. We did try to add ViT experiments in time for the rebuttal response but found models are prohibitively slow to train on our limited hardware (single dataset experiments require 30+ models and >100 points of the evaluation). We *will add ViT experiments* to the final version.
>
> **Claims are Agnostic to Model and Pretraining Choice:** We strongly emphasize that our arguments are not specific to any architecture choice or pre-training strategy - the *only requirement is that the representations are **high quality***, e.g., disentangled and expressive. We’ve updated our language to emphasize this.
>
> In fact, we use ResNet50s trained with MoCoV2 and CLIP as it has already been established that these models provide sufficiently high-quality pretrained representations for transfer learning [1]. Indeed, our central claim, that simplicity bias and feature distortion must be mitigated to induce for safe and strong adaptation, is agnostic to the choice of underlying model, pre-training dataset size, or training paradigm, because ***adaptation protocols** are responsible for inducing susceptibility to these pitfalls on downstream dataset.* For example, without intervention, SGD is known to prefer simple features. *Therefore, FT will remain susceptible to inducing simplicity bias, even if the underlying pretrained model has been improved!* While we acknowledge that bigger and better models may be less inherently predisposed to such pitfalls on standard benchmarks, our findings and the proposed variants remain important as we *do not know **apriori** how susceptible an **arbitrary** target dataset is to these pitfalls.*
>
> [1] Fine-Tuning can Distort Pretrained Features and Underperform Out-of-Distribution. Ananya Kumar, Aditi Raghunathan, Robbie Jones, Tengyu Ma, Percy Liang. In ICLR 2022.

---

> > ### Author Response · Authors · 2022-11-14
> > **Response to Reviewer AzrZ (Part 2)**
> >
> > >“However, given that currently the term large-scale suggests bigger models and datasets, I am not confident I can agree the studied setting can be deemed large-scale.”
> >
> > In our experiments, we used ResNet50 representations pre-trained on Imagenet-1K *and large scale image-text data (CLIP is trained using **400 million** pairs).* In the interest of evaluating on larger models, as suggested, *we have added an additional experiment on CIFAR10 using a **ResNet-101 pre-trained with SimCLR**,* where our key claims continue to hold (Table 9, pg 16). We will add additional larger models (ViTs) in the final version.
> >
> > We also highlight here the extent of our evaluation: each model is evaluated not only for ID/OOD accuracy but also for *adversarial accuracy, corruption accuracy (20 different corruptions at 5 severities), anomaly detection over 7 different datasets, and calibration over 3 distribution shifts.* Furthermore, our considered datasets *represent low/medium/high levels* of required distortion and capture *different types of distribution shift* (standard vs. subpopulation). Given our careful experimental design and the fact that our claims are not specific to choice of particular architecture or dataset, we believe that our insights will be useful to the community.
> >
> > > From the manuscript, it is not very clear to me what motivated connecting the empirical findings about the disparities in the safety metrics with the notion of simplicity bias. Moreover, would the same findings hold in case different metrics and / or datasets / architectures were considered?
> >
> > Thanks for the question! While simplicity bias is known to underlie several problems in safe machine learning, *we are the first to look at simplicity bias in the context of designing safer adaptation protocols.* Moreover, given that simplicity bias arises from *properties of stochastic gradient descent*, we expect our findings will hold in cases of different metrics or dataset or architectures. We have updated Section 4 to make this connection more clear, and briefly reiterate it here.
> >
> > SGD-trained models are predisposed to learning simple, linearly-separable features, in lieu of more expressive, complex features. Indeed, [1] recently demonstrated that *this bias is **extreme*** and models can become *invariant to useful **complex** features.* However, in cases of distribution shift, corruptions or adversarial attacks, *simple features are often misleading* and DNNs will generalize poorly on these samples. Given that safe machine learning requires that models are robust under these very conditions, we look to “avoiding simplicity bias” as a complementary perspective when designing adaptation protocols.
> >
> > In our setting, high-quality, SSL *pre-trained models have learned complex, disentangled features.* However, it is still possible for adaptation *protocols to induce features and decision boundaries that rely upon **simple** features,* harming safety performance. We clearly demonstrate this possibility on the dominoes dataset, where models rely upon the simple MNIST digit, instead of the more expressive, CIFAR image, even though complex pre-trained representations are well-suited for this target dataset. While this is an extreme example, this behavior *remains a possibility for real datasets* and may harm safety performance. Indeed, when we test our proposed variants (which were shown to help mitigate this bias in a synthetic setting) *we do see that safety and generalization do improve.*
> >
> > [1] The Pitfalls of Simplicity Bias in Neural Networks. Harshay Shah, Kaustav Tamuly, Aditi Raghunathan, Prateek Jain, Praneeth Netrapalli. NeurIPS 2020.
> >
> > **Summary:** We have added an additional experiment on CIFAR10 using a larger, ResNet101 pretrained with SimCLR, as per the reviewer’s suggestion, to provide additional support for claims across model size and architecture. While we will continue to add more model/pretraining paradigms to the manuscript, we emphasize that our claims are not specific to the particular choice of pretraining paradigm.

---

> > > ### Author Response · Authors · 2022-11-14
> > > **Response to AzrZ (Part 3)**
> > >
> > > ***Responses to Questions about Clarity, Novelty and Reproducibility:***
> > >
> > > >Even though Table 3 indicates the strategies to mitigate simplicity bias were helpful to improve the safety metrics, it is not clear whether those improvements are in fact a consequence of attenuating simplicity bias in the studies cases.
> > >
> > > Thanks for the comment! The very fact that safety metrics were improved suggests that simplicity bias was mitigated to an extent! To generalize well under image corruptions and adversaries, models must learn ***decision boundaries that are supported by features that remain discriminative*** under such perturbations. Notably, the ***simple features*** preferred by SGD-trained models, such as background or color, ***do not induce such decision boundaries***, leading to unsafe behavior. We acknowledge however, it can be difficult to directly measure simplicity bias on real-datasets, as the division between simple and complex features is not always clear, and attributing decisions to certain features can be challenging.
> > >
> > > To this end, we took the reviewer’s suggestion for a proxy measure and ***computed the gap between the hardest and easiest classes on CIFAR10 and Living17*** for both modified and vanilla LP+FT protocols. (DomainNet’s evaluation dataset is heavily class imbalanced, so we do not include those results.) Results are shown below. Here, we see that the ***proposed LP+FT variants generally have a smaller “gap” than the vanilla LP+FT protocol***, suggesting that they are more effective at mitigating simplicity bias. Designing metrics for directly measuring simplicity bias on real-datasets is an interesting direction of future work, and we will expand upon this discussion in the final version.
> > >
> > > | Protocol  | Living17   | CIFAR10      |
> > > |-----------|------------|--------------|
> > > | lp+ft     |     0.1300 |       0.0873 |
> > > | souplp+ft |     0.1500 |       0.0840 |
> > > | udplp+ft  | **0.1200** | _0.0737_ |
> > > | vatlp+ft  |   _0.1267_ |   **0.0580** |
> > >
> > > >The manuscript also lacks clarity due to an excess of acronyms (some of them were not defined in the text prior to use, such as DNNs), tables font is too small, and the text presents several typos.
> > >
> > > Thank you for this comment and catching these typos! We’ve incorporated feedback and submitted a more readable revised version. In particular, we’ve broken Section 3 into two observations, and moved our discussion of simplicity bias to the subsequent section to improve clarity.

---

> > > > ### Comment · Reviewer_AzrZ · 2022-11-17
> > > > **Updating score after rebuttal**
> > > >
> > > > Dear authors, thank you for the detailed response to my comments. I'm glad to see the empirical findings from the initial version of the manuscript to also hold in other scenarios. I also appreciate the reported proxy measure for simplicity bias and, as the results further support the empirical evidence in the work, I'm now more confident about the generalization and scalability of the reported findings, as well as the source of the improvements (i.e. there is stronger signal that simplicity bias is in fact being mitigated).
> > > >
> > > > Given that my concerns regarding empirical validation of the claims and clarity were all tackled in the rebuttal, I'm happy to raise my score from 5 to 8.

---

### Official Review · Reviewer_WDPP · 2022-10-25

**Confidence:** 4
**Correctness:** 4
**Technical Novelty And Significance:** 3
**Empirical Novelty And Significance:** 3
**Recommendation:** 6

**Clarity, Quality, Novelty And Reproducibility:**

The paper in general is a bit hard to read. I'd recommend clarifying the contributions + goal in the first half and then re-organizing section 3. The results are novel as it considers the role of simplicity bias in the context of model adaptation. See other sections for additional details.

**Strength And Weaknesses:**

Strengths:

1. Thorough empirical evaluation. The paper considers multiple variants of fine-tuning / adaptation, multiple evaluation metrics to measure model reliability in multiple axes, and multiple real-world and semi-real datasets.

2. Interesting perspective on model adaptation. The finding that simplicity bias (previously studied in trained-from-scratch) settings also plays an important role in the pretrain-and-finetune paradigm is insightful.

Weaknesses:

1. Hard to read. The paper (especially first few sections) lacks focus. I don't know what exactly the paper is trying to do (and what the main contributions are) even after re-reading the first 2-3 sections.

2. Empirical sections need re-organization. There is a lot going on in Section 3. It looks into (a) feature distortion insufficient to explain reliable adaptation, (b) motivates simplicity bias and (c) discusses the effect of LP initialization to mitigate simplicity bias with just one table. I would break this up into multiple smaller experiments to clearly showcase the findings.

3. Train from scratch baseline. An important but missing worst-case baseline for simplicity bias is if you train from scratch directly on the downstream task (i.e., without adaptation).

4. Role of LP initialization in mitigating simplicity bias understudied. I think the paper lacks concrete experiments that show that LP initialization is a major source of simplicity bias, given that it motivates the hardness-promoting variants in the later section(s).

5. (Minor) CKA unreliable as a metric. There are multiple papers that show that representation similarity metrics have failure modes (https://arxiv.org/abs/2108.01661, https://openreview.net/forum?id=8HRvyxc606). Using multiple evaluation metrics for representation similarity is one way to sidestep this issue.

6. (Minor) LP+FT as good as LP+FT variants. The LP+FT variants do not consistently improve model reliability in practice even thought it improves Rand OOD accuracy on semi-synthetic datasets. Having a discussion on this discrepancy might be useful.



**Summary Of The Paper:**

This paper studies the role of feature distortion and simplicity bias in the context of fine-tuning pre-trained models. They evaluate fine-tuned models using multiple metrics for model robustness / reliability; these findings on CIFAR10, DomainNet, Living17 show that simplicity bias, in addition to feature distortion, is needed to understand whether models adapt reliably. The paper also considers semi-synthetic datasets (with known simple features) to showcase the role of simplicity bias in model adaptation. Then, the paper considers multiple variants of LP+FT in order to mitigate simplicity bias (to some extent).

**Summary Of The Review:**

Please see weaknesses listed in the previous section.

---

> ### Author Response · Authors · 2022-11-14
> **Response to Reviewer WDPP (Part 1)**
>
> Thank you for your insightful feedback! We are delighted that the empirical evaluation was found to be “thorough” and our simplicity bias perspective on model adaptation was considered “interesting” and “insightful.” Below, we address all questions and concerns.
>
> > “Hard to read. The paper (especially the first few sections) lacks focus.”
>
> Thank you for this feedback! We have updated the introduction and contributions so that our main contributions are clearer.  We have also clarified why we focus on improving the LP initialization as a mechanism for reducing simplicity bias. Here is a summary of the updated contributions:
>
> **Proposed Work.** In this work, we seek to understand the factors relevant to the design of adaption protocols that promote effective and safe generalization. We take the first step towards this aim by demonstrating limitations in existing LP, FT, and LP+FT protocols through an extensive, joint evaluation, and studying adaptation protocols through the complementary lens of avoiding simplicity bias. Our contributions can be summarized as follows:
>
> **Sec. 3: Joint Analysis of Adaptation Protocol Safety and Generalization.** We show that when adaptation protocols are evaluated with respect to *both ID/OOD generalization and safety, LP+FT **trails behind** LP or FT* on several safety metrics. This demonstrates that ***solely** mitigating feature distortion is not sufficient for safe generalization.* We also observe that keeping subsequent *FT close to LP solution is crucial* for the improved OOD generalization of LP+FT. This motivates us to focus on improving the LP initialization as a mechanism for improving both safety and OOD performance.
>
> **Sec. 4: Role of Simplicity Bias in (Unsafe) Adaptation.** To understand how protocols may induce safe adaptation, we study how different protocols avoid simplicity bias. While ***simplicity bias*** (Shah et al. 2020) has been shown to *underlie several problems in machine learning safety*, to the best of our knowledge, *we are the first to consider its role in adaptation settings.* We demonstrate that protocols must ***not only*** reduce distortion, but that avoiding simplicity bias is ***also necessary*** for effective adaptation.
>
> **Sec. 5: Improved Protocols for Mitigating Simplicity Bias and Distortion.** We propose three, simple modified LP+FT protocols that help *mitigate both simplicity bias and distortion.* In particular, we consider modifying the LP step with uncertainty-driven perturbations (Pagliardini et. 2022), virtual adversarial training (Miyato et al. 2017) and model-soups (Wortsman et al. 2022), as they are simple and effective strategies. *Across synthetic and real datasets, the **modified** protocols help improve **safety and generalization.***
>
> >“Empirical sections need re-organization. There is a lot going on in Section 3. I would break this up into multiple smaller experiments to clearly showcase the findings.”
>
> Thanks for the suggestion! We have *updated the paper so that each point is explained in a **dedicated sub-section*** (Obs 1: Mitigating Feature Distortion does not induce safe adaptation, and Obs 2: Linear Probing Solutions Matter) *with an accompanying **explanatory figure***, (Fig.2: Disparate Performance of Adaptation Protocols and Fig. 3: Dataset-Level Distortion). Moreover, we have moved the discussion of simplicity bias to the next section to improve the clarity of Section 3.

---

> > ### Author Response · Authors · 2022-11-14
> > **Additional Experiment on Role of LP (Part 2 Cont'd.)**
> >
> > > “Role of LP initialization in mitigating simplicity bias understudied. I think the paper lacks concrete experiments that show that LP initialization is a major source of simplicity bias, given that it motivates the hardness-promoting variants in the later section(s).”
> >
> > To provide further experimental verification that mitigations should be undertaken during LP, *we have updated the revised manuscript with the following **additional experiment** (Supp C.2, pg. 14).* In brief, we evaluate two additional protocols, on the synthetic dominoes dataset, where VAT and UDP are *applied only during the FT step*, e.g., LP+FT(VAT), and LP+FT(UDP).
> >
> > We report the Randomized OOD Accuracy for these additional protocols in Fig. 6 (Supp. C.2), and summarize our findings here. We see that, across three different correlation ratios, *FT variants **lose performance** with respect to the LP mitigation variants.* Notably, LP + FT(UDP) loses up to 4% performance with respect to LP (UDP) +FT. While performance drops are not as large for VAT, we nonetheless see that LP+FT(VAT) loses performance (~1%) with respect to LP(VAT) + FT.
> >
> > These results further support our conceptual argument that mitigation strategies must be undertaken during the LP step to *ensure that subsequent FT is in a direction that preserves complex features.* We did not include a FT soup variant as it would be *prohibitively expensive to create soups of entire models* (instead of classifiers); highlighting the computational efficiency of implementing mitigation strategies during the LP step. We note that applying mitigation strategies during FT, in addition to LP, may further improve performance, and we will add these variants in the final version.

---

> > > ### Author Response · Authors · 2022-11-14
> > > **Response to Reviewer WDPP (Part 3)**
> > >
> > > > "(Minor) LP+FT as good as LP+FT variants. The LP+FT variants do not consistently improve model reliability in practice even though it improves Rand OOD accuracy on semi-synthetic datasets. Having a discussion on this discrepancy might be useful."
> > >
> > > Thanks for the comment! On real datasets, it is more challenging to distinguish between simple and complex features, which can make it difficult to see large benefits since the vanilla solution is well-suited for these particular datasets. *Since we cannot know a priori if the vanilla solution will be sufficient for arbitrary datasets, our variants help serve as a **preemptive protection** against this pitfall.* Moreover, while we selected UDP/VAT/model-soups for their effectiveness and ease-of-use, the design of more effective mitigation strategies remains an active direction of research. Indeed, as the quality of such mitigation approaches improves, we fully expect that *updating the LP step with these improved strategies will lead to larger and more consistent improvements.* We again emphasize that such strategies must be used *during the LP step* to ensure that subsequent FT is in a *direction that promotes learning **complex** features.* We have added this discussion to Sec 5. and intend to keep updating this manuscript in the future with improved mitigation strategies.

---

> > > > ### Comment · Reviewer_WDPP · 2022-11-19
> > > > **Post-rebuttal response**
> > > >
> > > > Thank you for the thorough response. The rebuttal and paper edits address my concerns / questions, so I am raising my score.

---

> > ### Author Response · Authors · 2022-11-18
> > **Response to Reviewer WDPP (Part 2)**
> >
> > > “Train from scratch baseline. An important but missing worst-case baseline for simplicity bias is if you train from scratch directly on the downstream task (i.e., without adaptation).”
> >
> > Thank you for the comment! In the updated manuscript, we have *included this baseline for the dominoes dataset* as it helps illustrate why adaptation given high-quality, pretrained models is helpful (Table 2). We see that the train-from-scratch baseline *obtains significantly poorer Randomized OOD Accuracy than **adapted** models.* Indeed, the near-random performance of the train-from-scratch baseline indicates that the model has failed to learn almost any complex features! In contrast, the better Randomized OOD Accuracy of adapted models indicates that these models are able to utilize the complex features from the pre-trained model to improve performance. While we note that while both settings are clearly susceptible to simplicity bias, well-designed adaptation protocols helped avoid this pitfall.
> >
> > We did not include this train-from-scratch baseline for CIFAR10/Living17/DomainNet as [1] already confirmed that *this baseline obtains significantly worse ID/OOD performance than **FT and LP.*** For example, on Living17, they find that the train-from-scratch baseline is “5% worse on ID” and “20% worse on OOD.”  Given that this baseline struggles on the clean data-distribution, it is highly *unlikely that it will be able to perform well on more **challenging** corrupted or adversarial distributions.* We will add this baseline to the final version for completeness.
> >
> > [1] Fine-Tuning can Distort Pretrained Features and Underperform Out-of-Distribution. Ananya Kumar, Aditi Raghunathan, Robbie Jones, Tengyu Ma, Percy Liang. In ICLR 2022.
> >
> > > “Role of LP initialization in mitigating simplicity bias understudied. I think the paper lacks concrete experiments that show that LP initialization is a major source of simplicity bias, given that it motivates the hardness-promoting variants in the later section(s).”
> >
> > Thanks for the feedback! We focus on the LP initialization to mitigate simplicity bias as we observed in Section 3 that LP+FT leads to *solutions that are in very close vicinity to LP solutions* (as evidenced by high CKA scores for Living17/DomainNet). Indeed, to see improvements in ID/OOD generalization over the LP solution for these datasets *it is essential to use **small** learning rates (3e-7, 1e-5) and **frozen** batch norm parameters during subsequent FT.* This observation indicates that a strong LP solution is critical for datasets where the subsequent FT step does not substantially change the solution. Moreover, in the case of extreme simplicity bias, we note that *interventions necessarily must be undertaken during LP.* Indeed, if the LP classifier relies upon simple features to find a low loss solution, during the subsequent FT, gradients *may not be back propagated in directions that contain **complex** features.* This entails that the decision boundary continues to rely upon simple features and remains brittle. For this reason, including mitigation strategies during FT and LP will likely further improve performance, but performing these strategies during the FT step alone will not be as effective in extreme bias settings. We note that by using mitigation strategies in the LP step, we are able to *reduce computational complexity* as we only need to create perturbations with respect to hidden representations and create soups of classifiers (instead of fully models).
> >
> > ***Summary:*** We have updated the manuscript to improve readability and clarified some points as per the reviewer’s helpful comments. We also added a train-from-scratch baseline to further illustrate the benefits of simplicity bias aware adaptation protocols.
> >
> > > (Minor) CKA unreliable as a metric. There are multiple papers that show that representation similarity metrics have failure modes. Using multiple evaluation metrics for representation similarity is one way to sidestep this issue.
> >
> > Thanks for the references! We have updated the manuscript to cite these papers, and added a note that other representation similarity metrics can also be used. We’re happy to add additional metrics for the final version.

---

### Author Response · Authors · 2022-11-18
**Summary of Updates**

We thank the reviewers for their thoughtful and constructive feedback! We are pleased they found the work to “tackle a relevant problem for the community”, provide a “thorough empirical evaluation”, and our investigation to be “insightful” and “extensive.” Here, we briefly summarize the updates made to the manuscript in response to reviewer feedback. (Changes are shown in blue. Headings in lieu of text are highlighted in the manuscript for readability.)

***Text Changes:*** The following updates have been made to improve readability.

- **Sec 1:** We have updated the “proposed work” section so that the contributions are clearer. We have also refined the introduction of different protocols and feature distortion.

- **Sec 3:** We have restructured this section into two observations (Observation 1: Mitigating Feature Distortion Does Not Induce Safe Adaptation and Observation 2: Linear Probing Solutions Matter), each with an accompanying figure (Fig 2: Disparate Performance of Adaptation Protocols and Fig. 3: Dataset Distortion). We also updated the language in this section so that it is more clear why the LP step is important.

- **Sec 4:** We have expanded the discussion in this section to better motivate why simplicity bias is an important and complementary perspective to feature distortion. We have also clarified why we use mitigation strategies in the LP step.

- **Sec 5:** We have expanded our discussion on the generality of our claims, namely that our central arguments are not dependent on the choice of mitigation strategies, architecture or pretraining paradigm.

***Additional Experiments:*** The following experiments have been conducted in response to reviewer comments.

- **Table 9: CIFAR10 with Resnet101/SimCLR Pretrained Model:** To demonstrate that our claims are supported on different architectures and larger models, we conducted an additional experiment with a ResNet-101 that is pretrained with SimCLR. Our proposed LP+FT variants continued to provide some safety and generalization benefits in this setting.

- **Figure 6: Applying Mitigation Strategies to FT:** To demonstrate that mitigations strategies should be applied in the LP step, we conduct an additional experiment where mitigations are applied to the FT step. We find that FT variants have worse performance than their LP counterparts.

- **Generalization Gap Proxy Measure:** As per Reviewer AzrZ’s suggestion, we computed the gap between the hardest and easiest classes on CIFAR10 and Living17 for both modified and vanilla LP+FT protocols. We found that our proposed variants generally have a smaller “gap” than the vanilla protocol, suggesting that they are more effective at mitigating simplicity bias.

***Additional Supplementary Discussion:*** The following discussions have been added to the supplementary in response to reviewer questions.

- **Supp. C1: Motivation for Hardness Promoting Variants:** We have included discussion on why UDP/VAT/Model-Soups were selected as simplicity bias mitigation strategies.

- **Supp C2: Applying Simplicity Bias Mitigation Strategies to Fine-Tuning Step:** We have included an additional experiment (Fig. 6) and discussion that explain the importance of improving the LP step when attempting to mitigate simplicity bias during model adaptation.

---

### Decision · Program_Chairs · 2023-01-20

**Decision:**

Accept: notable-top-25%

**Justification For Why Not Higher Score:**

There are some concerns on the clarity of the work.

**Justification For Why Not Lower Score:**

1. The research problem on model adaptation is important for the community.
2. The study is insightful. The evaluation is thorough and effectively supports the paper’s claims.


**Metareview: Summary, Strengths And Weaknesses:**

Summary:
The paper studies the role of feature distortion and simplicity bias in model adaptation with additional safety metrics (calibration, robustness, ID/OOD generalization). The proposed modified LP+FT protocols decrease the simplicity bias, and improve on OOD generalization and the additional metrics.

Strength:
1. The research problem on model adaptation is important for the community.
2. The study is insightful. The evaluation is thorough and effectively supports the paper’s claims.

Weakness:
1. There are few concerns on the clarity of the work.
2. [Minor] "safety" is an overloaded/confusing word. I wonder whether robustness or others will be more appropriate.



**Note From Pc:**

if the above contains the word "oral" or "spotlight" please see: "oral" presentation means -> notable-top-5% and "spotlight" means -> notable-top-25%. As stated in our emails, we are disassociating presentation type from AC recommendations